# Dual-wavelength metalens enables Epi-fluorescence detection from single molecules

Aleksandr Barulin [1,2,7], Yeseul Kim[3,7], Dong Kyo Oh [3,7], Jaehyuck Jang [4,5], Hyemi Park [1,2], Junsuk Rho [3,4,5,6] ✉ & Inki Kim [1,2] ✉

Single molecule fluorescence spectroscopy is at the heart of molecular biophysics research and the most sensitive biosensing assays. The growing demand for precision medicine and environmental monitoring requires the creation of miniaturized and portable sensing platforms. However, the need for highly sophisticated objective lenses has precluded the development of single molecule detection systems for truly portable devices. Here, we propose a dielectric metalens device of submicrometer thickness to excite and collect light from fluorescent molecules instead of an objective lens. The high numerical aperture, high focusing efficiency, and dual-wavelength operation of the metalens enable the implementation of fluorescence correlation spectroscopy with a single Alexa 647 molecule in the focal volume. Moreover, the metalens enables real-time monitoring of individual fluorescent nanoparticle transitions and identification of hydrodynamic diameters ranging from a few to hundreds of nanometers. This advancement in sensitivity extends the application of the metalens technology to ultracompact single-molecule sensors.

Single molecule sensing technologies are paramount for biosensing[1–3], single molecule chemistry[4–6], molecular dynamics[7,8], DNA sequencing[9,10], and proteomics[11]. Ultrasensitive fluorescence techniques can probe individual processes that would require synchronization in ensemble measurements[12]. In particular, fluorescence correlation spectroscopy (FCS) can gather dynamic data from multiple repetitive single molecule events. Although single molecule fluorescence techniques have advanced in the last three decades[13], the required equipment includes the most complex, costly, and bulky objective lenses in the market that make certain applications restrictive such as point-of-care diagnostic settings and environmental monitoring. Previously, miniaturized smartphone-based microscopes were reported to enable single nanoparticle detection with low numerical aperture (NA) optics[14–16], while integrating them with an addressable nanoantenna platform can push the sensitivity to single molecule level[17]. Alternatively, hand-held lensfree holographic imaging platforms could provide single nanoparticle observation above 40 nm[18]. Conversely, dielectric metalenses enable unprecedented electromagnetic field tailoring to focus light close to the diffraction limit with high NA[19–23], whereas their micrometer-thick arrangement of meta-atoms is suitable for miniaturized imaging systems[24]. Metalens studies have advanced toward chromatic aberration correction at two or three wavelengths or within a broad bandwidth[25–30]. The compactness and focusing performance make the metalens a suitable candidate to further miniaturize single molecule detection systems toward truly portable devices[16]. Dielectric metasurfaces operating as phase and polarization masks in the microscope Fourier plane are able to get rid of localization biases in single-molecule imaging[31]. However,

[1]Department of Biophysics, Institute of Quantum Biophysics, Sungkyunkwan University, Suwon 16419, Republic of Korea. [2]Department of Intelligent Precision Healthcare Convergence, Sungkyunkwan University, Suwon 16419, Republic of Korea. [3]Department of Mechanical Engineering, Pohang University of Science and Technology (POSTECH), Pohang 37673, Republic of Korea. [4]Department of Chemical Engineering, Pohang University of Science and Technology (POSTECH), Pohang 37673, Republic of Korea. [5]POSCO-POSTECH-RIST Convergence Research Center for Flat Optics and Metaphotonics, Pohang 37673, Republic of Korea. [6]National Institute of Nanomaterials Technology (NINT), Pohang 37673, Republic of Korea. [7]These authors contributed equally: Aleksandr Barulin, Yeseul Kim, Dong Kyo Oh. ✉e-mail: jsrho@postech.ac.kr; inki.kim@skku.edu

attaining a single molecule level of fluorescence sensitivity under the epi-fluorescence operation implies that the lens simultaneously possesses high focusing efficiency, high NA, and achromatic focusing. Finding a trade-off between these functionalities in a single metalens device poses a fundamental challenge[32,33]. In contrast, near-unity NA metalenses have previously been demonstrated to detect the emission of individual diamond nitrogen-vacancy (NV) centers[21,22]. However, these metalenses only operate as one-wavelength collecting lenses, which hinders their realistic miniaturization and application as portable devices for epi-fluorescence detection. Moreover, NV centers are typically brighter than conventional organic fluorescent molecules by two or three orders of magnitude[34,35] and yield extraordinary photostability, drastically relaxing the requirements for metalens performance. In contrast, dual-wavelength metalenses have been shown to enable two-photon fluorescence imaging of large microparticles in the epi-fluorescence mode[36]. However, single molecule or single nanoparticle detection by a metalens has remained elusive.

In this study, we address the challenge of metalens fluorescence sensitivity and utilize FCS to capture fluorescence fluctuations arising from single molecule and single nanoparticle passes driven by 3D-Brownian motion in an aqueous solution. The FCS technique can directly track single-molecule diffusion events and provide information hidden from ensemble-averaging techniques, including the number of molecules in the detection volume, molecule brightness, and hydrodynamic radius, even in the case of low optical system collection efficiency or reduced emission quantum yield[37–39]. We designed a metalens made of hydrogen-doped amorphous silicon (aSi:H) meta-atoms[40]. The metalens possesses several functionalities simultaneously, i.e., high NA of 0.6, high focusing efficiency, and chromatic aberration correction at two wavelengths. In contrast, the metalens operates in one-photon epi-fluorescence mode without an objective lens to excite and collect fluorescent light from diffusing emitters in an aqueous solution. The confocal configuration and nanosecond signal time gating allowed us to detect fluorescence from the focal volume of the metalens with modest background noise and out-of-focus fluorescence. We employ Alexa Fluor 647 as a benchmark fluorophore commonly used in bioimaging and biosensing techniques, and it yields highly photostable conjugates and optimized blinking behavior under red light excitation[41–43]. Additionally, Alexa 647 has emerged as a standard fluorophore in conventional FCS studies in aqueous solutions and living cells[44,45]. The designed metalens provides a high enough sensitivity to capture the fluorescence signal and FCS correlation function from 1.7 Alexa 647 molecules in the detection volume. Compared to conventional objective lenses with similar properties, the designed metalens yields comparable single molecule sensitivity with a single layer of dielectric nanofins, while the lens size is miniaturized by around 2 orders of magnitude in all dimensions. Furthermore, we monitor the real-time fluorescence bursts of quantum dots (QDs) and stained nanoparticles (NPs). The nanoparticles yield emission spectra that partly overlap with that of Alexa 647 and, hence, appear compatible with the metalens platform. NPs with different hydrodynamic diameters are clearly distinguished based on their diffusion time through the metalens focal volume. This unprecedented sensitivity opens up various metalens applications in portable single-molecule systems. Because of their small size, metalenses can be integrated with available miniaturized microscope solutions, *e.g.*, smartphone microscopes[16,17] and 3D-printed FCS platforms[46]. Metalens structures are compatible with silicon photonic chips[47,48] making them viable candidates for integrating single molecule on-chip sensors for point-of-care testing[1].

## Results
### Design principle of dual-wavelength metalens
Figure 1a shows the operational scheme of the metalens for the excitation and collection of the fluorescence of diffusing single Alexa Fluor

647 molecules. The two working wavelengths of 635 and 670 nm correspond to the laser line of our microscope and the emission maximum of Alexa 647 molecules (Fig. 1b). The rationale behind selecting these target wavelengths is to ensure efficient fluorescence collection from the laser spot and avoid excessive collection of out-of-focus uncorrelated background light. Single molecule spectroscopy experiments are conducted using a home-built time-resolved confocal epi-fluorescence microscope (Fig. 1c). The spectral filter set is optimized for fluorescence detection in the spectral band between 655 and 700 nm (see details in "Methods"). The metalens focuses laser beam through a thin glass slide inside a fluorescent solution. The designed metalens device substitutes the excitation and collection objective lens on the microscope body.

The metalens is composed of rectangular meta-atoms made of aSi:H that exhibit a high refractive index ($n$) and a low extinction coefficient ($k$) at the working wavelengths (Supplementary Fig. 1). The diameter of the metalens is 500 μm, and the focal length ($f$) is 330 μm (NA = 0.6). The phase maps of the dual-wavelength metalens are calculated by satisfying the lens equation: $\varphi(x,y) = 2\pi/\lambda \cdot (f - \sqrt{f^2 + x^2 + y^2})$, where $\varphi$ is phase, $x$ and $y$ are the spatial coordinates, respectively, and $\lambda$ is the target wavelength (Supplementary Fig. 2). Two theoretically ideal phase maps can be produced as there are two target wavelengths. The metasurface operates on the principle of phase modulation using the co-polarization term of the propagation phase (Supplementary Fig. 3)[49]. By exploiting the discrepancy in transmittance between the major and minor axes, it becomes possible to implement a phase variation ranging from 0 to 2π. Furthermore, this metasurface possesses polarization-independent characteristics, enabling it to focus the linearly polarized laser beam and collimate the emitted nonpolarized fluorescence light. Through simulations using the rigorous coupled wave analysis method[50], we obtain the values of transmittance, conversion efficiency, and phase while sweeping the width (W) and length (L) from 80 to 280 nm at 5 nm intervals (Supplementary Fig. 4), with a periodicity (P) of 300 nm and a height (H) of 500 nm.

Consequently, based on the simulation results, we selected structures exhibiting transmittances above 50% and conversion efficiencies exceeding 50%. Among these structures, we identify the nanostructure geometry at each position that best approximated the theoretical ideal phase map for the incident wavelengths of 635 and 670 nm, achieving a matched phase map (Supplementary Fig. 2). Supplementary Fig. 5 shows the values of W and L for the nanostructures in the matched phase maps, respectively. The transmittance and conversion efficiency maps are shown in Supplementary Fig. 6. Under the matching conditions, the average transmittance of the obtained metalens amounts to 82.6% at 635 nm and 83.5% at 670 nm, whereas the average conversion efficiency was 71.6% at 635 nm and 76.2% at 670 nm. This result represents the conversion efficiency obtained when using the co-polarization term, which is more than 5% higher than the conversion efficiency observed when using cross-polarization (Supplementary Fig. 7). Next, the metalens is fabricated on a glass substrate using electron-beam lithography ("Methods"). A representative optical microscopy image of the metalens is shown in Fig. 2a. According to the scanning electron microscope images, the fabricated metalens accommodates well-defined rectangular meta-atoms with sharp edges (Fig. 2b, c). The rationale behind the optimization of focusing efficiency, NA, and chromatic aberration correction takes into consideration two main aspects: overlap of excitation and detection volumes (Fig. 2d) and molecule detection efficiency (MDE). The overlap mismatch of the excitation and detection volumes leads to the elongation of the FCS effective volume and limited single molecule brightness even for high NA lenses[51]. MDE represents a metric of single molecule fluorescence intensity the system can detect and amounts to a product of the laser power density, i.e. the rate of pumping the

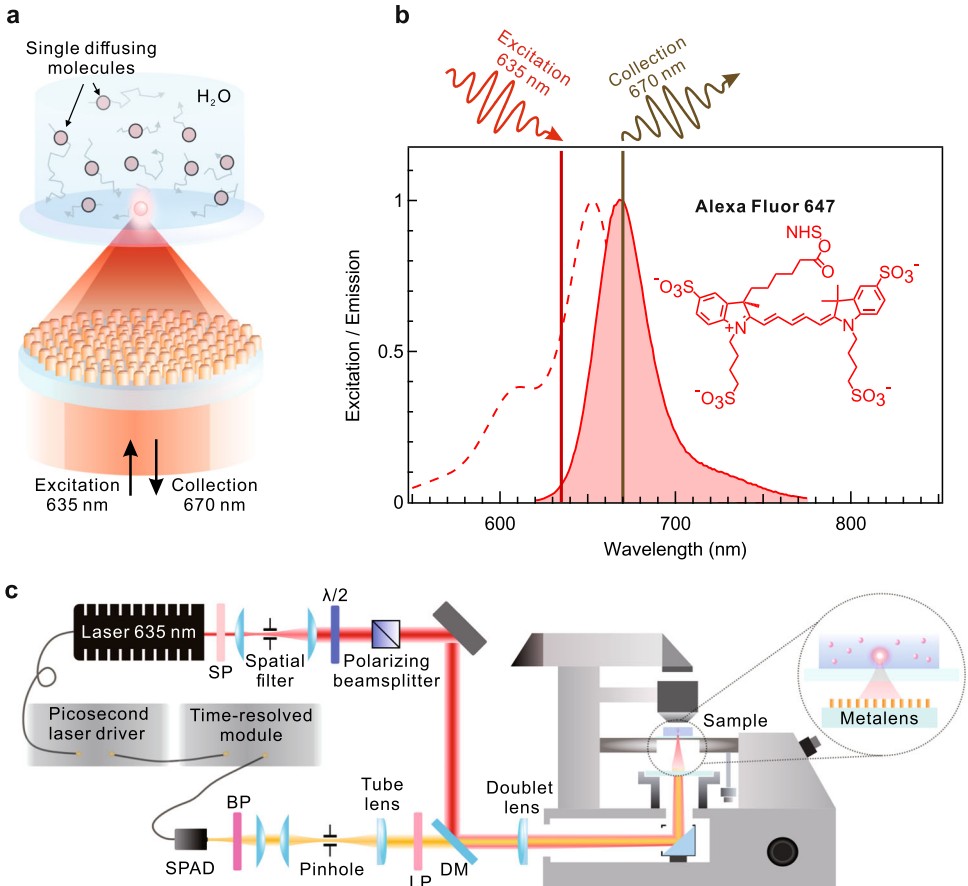

**Fig. 1 | Dual-wavelength metalens for single molecule detection. a** Schematic of proposed concept. Metalens focuses 635 nm laser inside fluorescent molecule solution and collects fluorescence light at 670 nm. **b** Fluorescence spectrum of Alexa 647 molecule. Vertical lines mark target wavelengths of metalens corresponding to laser line and emission maximum of Alexa 647. **c** Confocal time-resolved setup with integrated metalens. SP shortpass filter, λ/2 half-waveplate, DM dichroic mirror, LP longpass filter, BP bandpass filter, SPAD single photon avalanche diode.

molecules to their singlet excited state, and the collection efficiency of the system (CEF)[52]. Under the diffraction limit conditions, the laser power density is proportional to $FE_{exc} \cdot NA^2$ with $FE_{exc}$ being the metalens focusing efficiency (transmittance in case of conventional objective lens) at the laser wavelength. On the other hand, for a correctly set pinhole size, CEF is proportional to $FE_{em}(1 - \sqrt{1 - (NA/n)^2})$ [52], where $FE_{em}$ is the metalens focusing efficiency (transmittance in case of conventional objective lens) at the emission wavelength, n is the refractive index of the aqueous solution. Figure 2e displays the MDE plot as a function of NA at four exemplary metalens focusing efficiencies. Both NA and focusing efficiency drastically affect MDE, therefore, our rationale for a fabricable metalens design accommodates their trade-off together with dual-wavelength operation.

## Evaluation of focusing performance

Focusing tests are conducted at the two target wavelengths to evaluate the performance of the fabricated metalens. The focal spots of the metalens are characterized using a transmission microscope (Supplementary Fig. 8) with laser wavelengths of 635 and 670 nm. Figure 3a, b shows the resulting focal spot intensity images. The focal spots exhibit a highly circular shape (Fig. 3c) with a full width at half maximum (FWHM) of 0.77 µm at 635 nm and 1.03 µm at 670 nm. Figure 3d represents the scanning results in the z-direction used to evaluate the depth of focus. The experimental focal spots are enlarged because of several reasons including phase mismatch and manufacturing imperfection (see details in Supplementary Note 9).

Therefore, the experimentally measured NA is the effective NA, which is determined by physical diameter and focal length. The images confirm that the focal points are positioned at a distance of 330 µm corresponding to effective NA = 0.6, which is large enough to focus laser light through a thin glass slide. The focal distances match nearly perfectly, and the metalens focal volume aspect ratio is approximately 10. We define the focusing efficiency as the ratio of the light intensity captured at the focal spot to the amount of incident light entering the metalens. At the incident wavelength of 635 nm, the focusing efficiency is measured to be 14.2%, whereas, at 670 nm, it increases to 37.1%. The higher focusing efficiency at 670 nm as compared with that at 635 nm is attributed to the shorter depth of focus. Compared with the case in which the theoretically ideal phase is applied (Supplementary Fig. 11), the FWHM increases 1.35 and 1.89 times for the respective wavelengths. In addition, the depth of focus has increased, which can be attributed to two factors: (i) the challenge in achieving a perfectly identical phase with the matched phase approach and (ii) the difficulty in patterning nanostructures into perfect rectangular prisms owing to fabrication limitations. Nevertheless, the desired properties have not been achieved with single layer metalenses before, as we conduct the metalens performance comparison to achromatic lenses from prior arts (Supplementary Table 1). As creating multilayer metalenses with several layers is typically accompanied by layer alignment issues, the advantage of the metalens design of this work is implementation of single layer metalens with tight focusing and sufficiently high focusing efficiency at the fluorophore emission wavelength.

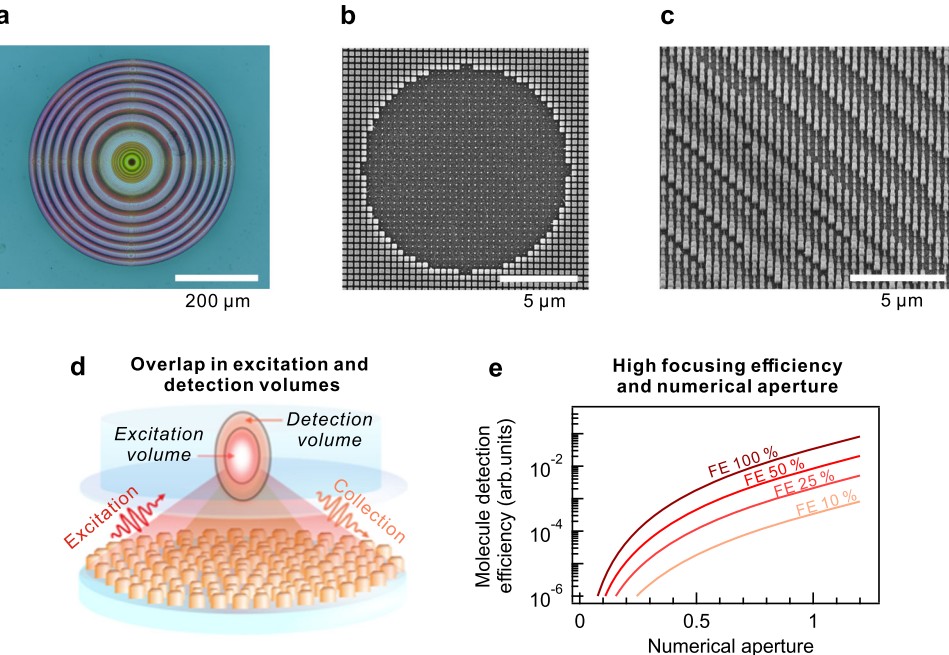

**Fig. 2 | Metalens design. a** Bright-field optical microscope image of fabricated metalens. **b, c** Scanning electron microscope images of metalens in center and away from center, respectively. **d** Schematic of dual-wavelength metalens functionality. Overlap between excitation and detection volume minimizes confocal effective volume and provides efficient light collection. **e** Calculated molecule detection efficiency as function of lens NA and focusing efficiency (FE). Molecule detection efficiency is represented in arbitrary units, as its absolute values strictly depend on total transmission of optical microscope setup. FEs at excitation and collection wavelengths are considered identical for this schematic.

## Metalens FCS of single molecules

Afterward, the metalens is mounted onto a confocal epi-fluorescence microscope for single-molecule fluorescence sensing. We incorporate an achromatic doublet lens to adjust the beam width to the size of the metalens. The effective focal length of the metalens-doublet lens system defines the resulting ×31 magnification of the metalens microscope (see detailed calculations in Methods). Considering the metalens point spread function (PSF) data and magnification, we employed an 80 μm confocal pinhole. We record the fluorescence intensity time traces of the diffusing Alexa 647 molecules in a phosphate buffer solution (PBS, pH 7.4) (Fig. 4a). The laser power (1 mW) is below the fluorescence saturation intensity of the molecules (Supplementary Fig. 12). We speculate that a high focusing efficiency at a collection wavelength of 670 nm brings more value to the metalens sensitivity than that at the excitation wavelength, as the molecule photon budget is intrinsically limited[53]. The time traces are acquired at concentrations from 6 nM to 10 pM for approximately 10 min to accumulate a sufficient amount of single molecule events and reduce the FCS correlation function noise. The time traces are filtered using a post-processing time-gating technique[54,55] in a 10 ns fluorescence decay window. Time-gating filtering minimizes the contribution of the long-lived metalens background noise. The possibility of recording the FCS correlation functions of the correct amplitude and decay time validates the single molecule sensitivity of the metalens system. We confirm that the correlation is zero if the metalens is defocused from the fluorescent molecule solution or completely removed from the light path (Supplementary Fig. 13). Furthermore, the correlation amplitudes increase as the number of molecules decreases (Fig. 4b). We fit the FCS correlation functions by the 3D-Brownian diffusion model (see Methods) with the fixed beam aspect ratio κ = 10 and extract the relevant parameters such as molecule diffusion time, number of molecules in the confocal effective volume of the metalens, single molecule brightness (CRM). The diffusion time through the metalens focal volume resides at approximately 2.2 ms (Supplementary Table 2). The dark-state blinking term with a characteristic time of

5 μs for Alexa 647 (Supplementary Fig. 14) is not resolved because of the lack of single molecule brightness. Therefore, considering the standard diffusion model, we analyzed the autocorrelation functions with lag times above 10 μs. The collected non-diffracted fluorescence light which is emitted away from the focal volume affects the correlation amplitude at high concentrations. In contrast, the residual background noise of the metalens structure restrains the correlation amplitude to below approximately $15 \times 10^{-3}$ at low concentrations. To correctly identify the number of molecules from the FCS correlation function fits, we account for the background intensity, *i.e.*, the signal acquired when the metalens focuses the laser light below the fluorescent solution reservoir (see Methods). The time traces and FCS correlation functions can be recorded for only 1.7 molecules (Fig. 4c) at 10 pM concentrations, consolidating the ultimate sensitivity of the designed metalens. The number of detected molecules scales linearly with the concentration (*C*) from one to several thousand molecules. The apparent FCS detection volume in the aqueous solution amounts to 280 fl which exceeds the confocal effective volume determined from the PSF data by around 5 times (see details in Methods). We attribute this focal volume increase to the collection of a fraction of non-diffracted ballistic fluorescent photons by the metalens and focal spot aberrations induced by light propagation through glass slides and aqueous solutions. Moreover, remaining chromatic distortion can contribute to the effective volume expansion, as it has been clearly demonstrated on near-unity NA aspheric lenses[52]. We directly retrieve the single molecule brightness by dividing the fluorescence intensity collected from the detection volume by the number of molecules (Methods). The fluorescence intensity of a single molecule is approximately 50 counts/s. Next, we monitor the diffusion time delay of Alexa 647 molecules in a highly viscous solution (glycerol 60 vv% in PBS). From the observed correlation decay (Fig. 4d), the diffusion time in glycerol increases by approximately 14×, which is in good agreement with conventional objective lens measurements (Supplementary Fig. 14). Figure 4e shows the fluorescence decays of Alexa 647 determined by time-correlated single photon counting (TCSPC). Alexa 647

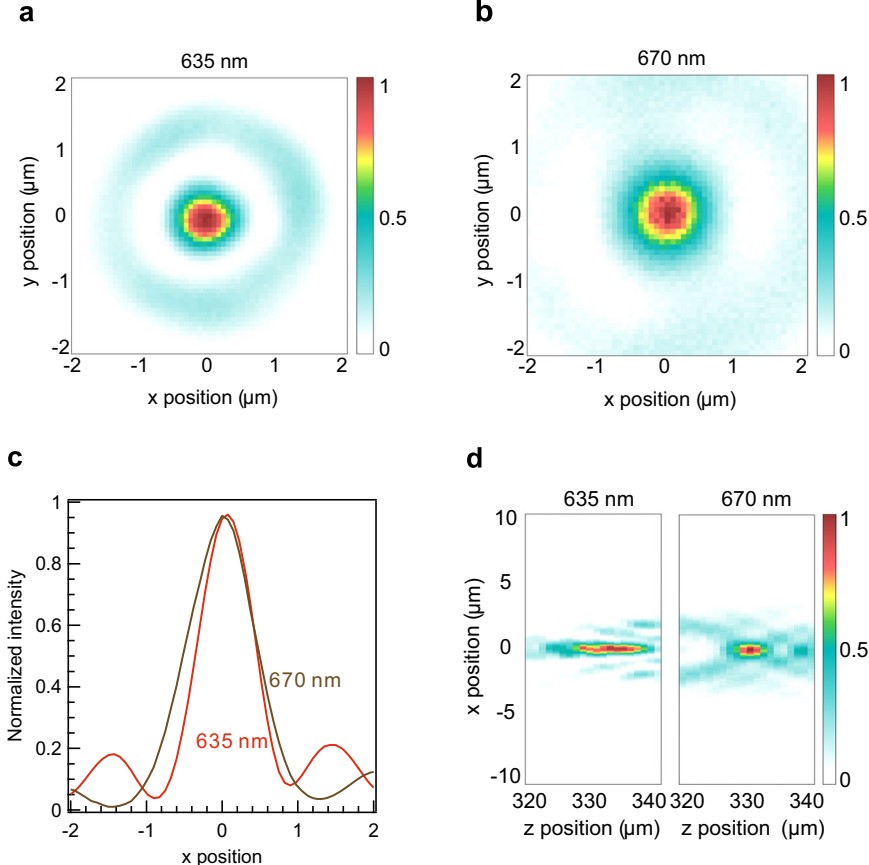

**Fig. 3 | Metalens focusing characterization. a, b** Lateral intensity profiles of metalens at 635 and 670 nm, respectively. **c** Corresponding horizontal cuts of (**a**) and (**b**). **d** Axial intensity profiles at 635 nm (left panel) and 670 nm (right panel). Scanning measurements in z-direction are carried out at 1-μm intervals.

yields natural lifetime values of 0.97 ns without glycerol (Supplementary Table 3). The mono-exponential decay and lifetime value agree with the expected Alexa 647 photophysical properties[56]. The Alexa 647 lifetime, which is sensitive to the glycerol environment[57] increases to 1.57 ns in the glycerol solution.

In order to provide an adequate comparison between our metalens performance and the conventional optics, we record FCS data of diffusing Alexa 647 molecules by a set of lenses of different NA and chromatic aberration presences (Fig. 5a). The fluorescence from Alexa 647 solution is excited and collected by two aspheric lenses of NA = 0.18 and 0.54 and objective lenses of various NA in the range of 0.15 and 1.2 (lens sizes shown in Supplementary Table 4). The corresponding raw FCS correlation traces are shown in Supplementary Fig. 15. We observe that the aspheric lenses of NA close to that of the metalens or below do not enable reaching single molecule sensitivity under identical excitation/detection conditions (Fig. 5b). The lack of sensitivity comes from the mismatch in overlap between the excitation and detection volume of the single-element aspheric lens due to its intrinsic chromatic distortion[51,52]. Employing achromatic multi-element objective lenses clearly boosts the single molecule fluorescence intensity (Fig. 5c), although it still remains undetectable for objective lenses with NA < 0.25. The designed metalens exceeds the performance of objective lenses with NA < 0.5 by all critical FCS parameters such as single molecule brightness, effective FCS volume, and diffusion time of molecules (Fig. 5c–e). Although the metalens-enabled single molecule brightness and diffusion time fall short of those retrieved by NA = 0.5 objective lens, the effective volume comes close to the conventional objective lens trend. Altogether, the proposed metalens device miniaturizes conventional objective lens size by 2 orders of magnitude in all dimensions and maintains similar single

molecule sensitivity. Moreover, the demonstrated dual-wavelength operation of the metalens encoded within a single nanostructure layer plays a pivotal role in providing sufficient collection efficiency from diffusing molecules, whereas conventional single-element refractive lenses typically strive to collect sufficient single molecule fluorescence signal even at high NA[52].

## Size identification and real-time monitoring of single nanoparticles

Single nanoparticle detection is relevant for environmental microplastic monitoring[58] and biomedical viral tests[14]. FCS allows us to extend the metalens applicability to the detection of different types of emitting single nanoparticles with the capability of size and concentration identification at nanometer precision. We perform FCS measurements on carboxyl quantum dots (QDs) with a hydrodynamic diameter ($D_h$) of 11 nm (emission maximum at 655 nm) and plastic NPs with $D_h = 110$ nm and $D_h = 490$ nm, which are stained with dyes emitting at 680 nm (Fig. 6a). The excitation power is reduced to 100 μW to avoid photobleaching of the NPs (Supplementary Fig. 12). The autocorrelation function demonstrates clear shifts toward longer emitter transit times as the hydrodynamic diameter of the nanoparticles increases. This size range lines up with nanoplastic size, i.e., microplastic below 1 μm that is released from biomedical care products, paints, or upon microplastic degradation[58] and obey Brownian motion interactions[59]. The metalens FCS can distinguish nanoparticle sizes far below the diffraction limit, where spatial analytical methods of microplastic are typically flawed[60]. The determined number of NPs follows in excellent agreement with the dilution of the colloidal solution (Supplementary Fig. 16). As expected from the Stokes–Einstein law, the determined diffusion time increased linearly with the

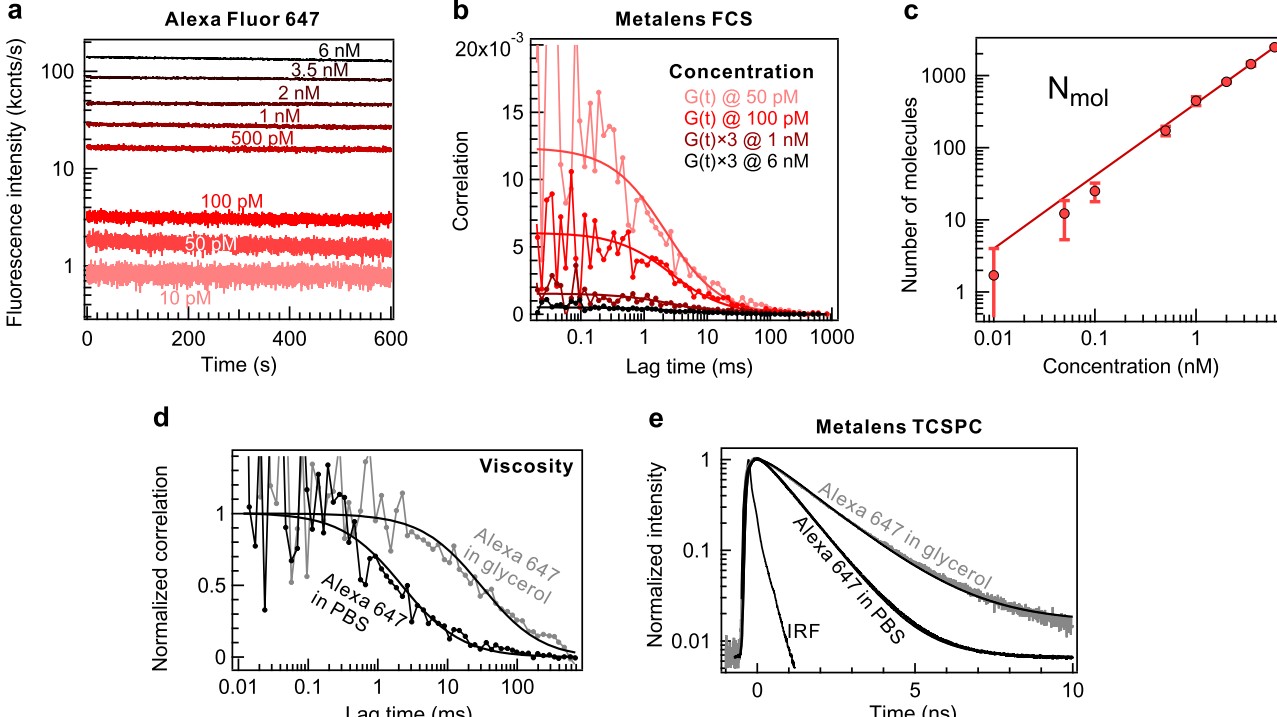

**Fig. 4 | Metalens-enabled single molecule fluorescence detection.**
**a** Fluorescence intensity time traces acquired at different concentrations of Alexa 647 in phosphate buffer solution (PBS). **b** Retrieved autocorrelation functions from selected time traces in Panel **a**. Autocorrelation functions at 1 and 6 nM are multiplied by 3 for clarity. **c** Number of molecules in metalens detection volume directly retrieved from FCS correlation functions and background intensity data. Error bars denote standard deviation of FCS experiment. **d** Normalized FCS correlation functions of Alexa 647 in the presence and absence of glycerol. Diffusion time in glycerol 60 vv% solution rises by approximately 14× owing to viscosity increase. **e** Fluorescence decay data of Alexa 647 in the absence and presence of glycerol 60 vv%.

hydrodynamic diameter (Fig. 6b). Similarly, the brightness per nanoparticle increases with the size, following the increasing absorption of the nanoparticles (Fig. 6c). The NPs 490 nm deliver bright and long fluorescent bursts of >10 kcounts/s. Furthermore, we dilute the colloidal solution of NPs 110 nm and QDs such that only one nanoparticle or less is present in the detection volume (Supplementary Fig. 17). Nanoparticle events produce bursts above the threshold value of the buffer solution as the nanoparticle brightness exceeds the shot noise of the Poisson statistics. The time trace histograms feature tails at higher count rates, even in the presence of approximately 0.17 NPs, on average, which could correspond solely to single nanoparticle events. Finally, the fluorescence origin of the signal from the NPs is confirmed by fluorescence decay traces (Fig. 6d), with widths substantially exceeding the instrument response function (IRF).

## Discussion

In conclusion, to the best of our knowledge, we have experimentally demonstrated that a flat metalens device enables detection of single fluorescent molecules without any objective lens. The epi-fluorescence operation simplifies and miniaturizes the metalens microscope apparatus. These results confirm that the metalens platform can compete with complex and costly objective lenses for single molecule sensing and dynamics studies. Observing temporal signal fluctuations provides access to molecular parameters, such as the molecular diffusion coefficient and concentration, which could extend to molecular interaction dynamics and conformational rearrangements. Moreover, we record single diffusing nanoparticles in real time and unambiguously identify emitter hydrodynamic diameters ranging from one to hundreds of nanometers using the FCS approach. We admit that there is still room for signal improvement by pushing the metalens design toward the ultimate theoretical limits of focusing efficiency, numerical

aperture, and broadband operation[61]. The demonstrated performance already allows considerable miniaturization and substitution of costly objective lenses of modest NA. Considering that future improvements in metalens fabrication and performance could achieve the single molecule sensitivity of current conventional state-of-the-art measurements and enable additional functionalities with unprecedented control of light properties. For instance, the unique tunability of electromagnetic field manipulation by metasurfaces[62–64] opens horizons for future developments in single molecule spectroscopy, such as directionality manipulation and sorting of emission or additional system miniaturization of multicolor detection systems by substituting functionalities of beam splitters and spectral filters. Another advantage of the proposed methodology includes its potential integration with miniaturized microscopes or photonic integrated circuits for handheld single molecule sensors with ultimate miniaturization, portability, and mass producibility[65].

## Methods

### Metalens fabrication and characterization
The dual-wavelength metalens is fabricated using a high-resolution electron beam lithography (EBL) system on a glass substrate. First, a high-refractive-index a-Si:H is deposited with 600-nm thickness on the substrate by a plasma-enhanced chemical vapor deposition at a specific condition with gas flows of about 10 sccm for $SiH_4$ and 75 sccm for $H_2$. After spin coating and soft baking a photoresist (Microchem, 950 PMMA A2) on the a-Si:H deposited substrate, the EBL system (ELIONIX, ELS-7800) is used to draw an optimized metalens design, followed by a development process using a methyl isobutyl ketone/Isopropyl alcohol 1:3 solution. In order to define an etch mask, A 40-nm thick chromium (Cr) is deposited on the patterned photoresist by electron beam evaporation system (KVT, KVE-ENS4004) and then the patterned

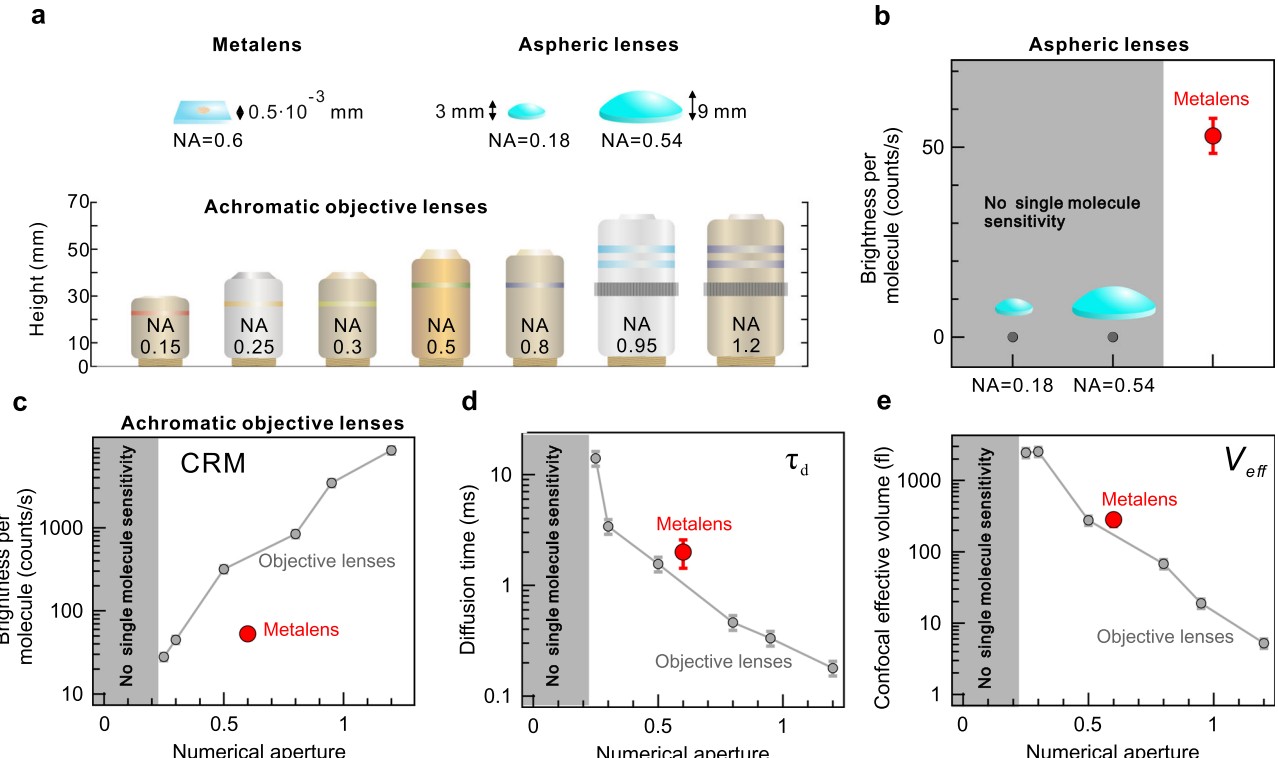

**Fig. 5 | Metalens performance comparison against conventional optics. a** Set of lenses employed for comparative study with indicated size dimensions: metalens, two single-element aspheric lenses of NA = 0.18 and NA = 0.54, and achromatic objective lenses of NA between 0.15 and 1.2. **b** Alexa 647 single molecule brightness (CRM) detected by metalens and aspheric lenses. As FCS correlation is not detectable by aspheric lenses, single molecule brightness is considered to equal zero. **c** Alexa 647 single molecule brightness detected by metalens and achromatic objective lenses. **d** Alexa 647 molecule diffusion time ($\tau_d$) detected by metalens and achromatic objective lenses. **e** Confocal effective volume ($V_{eff}$) observed by metalens and achromatic objective lenses. Error bars denote standard deviation of FCS experiment.

photoresist is completely removed by acetone, leaving only Cr masks on the substrate for a selective etching process of the a-Si:H layer. After the dry etching (DMS, silicon/metal hybrid etcher) removes residual a-Si:H area not protected by Cr masks, the final metalens is achieved by wet-etching of the remaining Cr masks using Cr etchant (CR-7). All scanning electron microscopy (SEM) images are obtained using a field-emission SEM system (HITACHI, Regulus 8100) with typical operating voltage of 5kV.

The focal spots of the metalens are characterized using a camera-based microscope. Lasers with working wavelengths of 635 and 670 nm are used to produce spatially overlapping beams of light that are then adjusted by two lenses with the fiber coupling system to make the incident beam Gaussian. Two dichroic mirrors are employed to propagate the three beams at normal incidence. The metalens focal spots are magnified using a scanning horizontal microscope with an objective lens (Olympus PLANFL 100x, NA 0.8), a tube lens (Thorlabs TTL180-A), and an sCMOS camera (PCO panda 4.2). The horizontal microscope is mounted on a motorized stage (Thorlabs DDS100) to measure the intensity of the light with 1-μm intervals.

**Time-resolved confocal microscope with metalens**
The excitation source is a 635-nm picosecond laser (PicoQuant, LDH P-C-635) with a pulse width of approximately 120 ps. The laser repetition rate and synchronization are driven by the laser driver (PicoQuant, Sepia PDL 828). The laser beam is spatially filtered to produce a quasi-Gaussian profile. The possible long-wavelength light emitted by the laser is filtered out using a short-pass filter (Thorlabs, FESH0650). A dichroic mirror (Semrock, FF649-Di01-25 × 36) splits the excitation and fluorescence light. The laser power values in the text correspond to the incident power on the dichroic mirror. The incident power is adjusted using a half-waveplate placed before the polarizing beam splitter. A doublet lens ($f$ = 200 mm, Thorlabs, AC254-200-AB) is used to adjust the size of the incident beam to the aperture of the metalens. The molecular fluorescence is excited and collected using dual-wavelength metalens mounted on a microscope body (Nikon Eclipse Ti2). The metalens and doublet lens are removed in conventional fluorescence correlation spectroscopy (FCS) control experiments, and the following lenses were employed: aspheric lenses (Thorlabs, AL2520-A; Thorlabs, C560TME) and objective lenses (Olympus MPlanFL N NA = 0.15, 5x; Olympus Plan N NA = 0.25, 10x; Olympus MPlanFL N NA = 0.3, 10x; Olympus UPlanFL N NA = 0.5, 20x; Olympus MPlanFL N NA = 0.8, 50x; Nikon PLAN APO λD NA = 0.95, 40x; Nikon Plan Apo VC, NA = 1.2, 60x, WI). The fluorescent light passes through a long-pass filter (Semrock, BLP01-635R-25), tube lens (Thorlabs, TTL200-A), confocal pinhole, and band-pass filter (Semrock, FF01-679/41-25) and is focused on a single photon avalanche diode (MPD, PD-050-CTC). The pinhole size is 80 μm in the case of the metalens measurements and 50 μm in the case of the objective lens control experiments. TCSPC and FCS data are recorded using a time-resolved module (PicoQuant, Multiharp 150 4 P) with time resolution of 5 ps.

The effective focal length of the metalens-doublet lens system is given by a relation of $1/f = 1/f_{meta} + 1/f_{DL} - \left(\frac{d}{f_{meta}} f_{DL}\right)$ where $f_{meta} = 0.33$ mm and $f_{DL} = 200$ mm are the focal lengths of the metalens and doublet lens, respectively, and $d$ is the distance between the metalens and doublet lens. The distance $d$ is 190 mm, which defines the effective focal length of the metalens-doublet lens system as 6.45 mm. Given that the focal length of the microscope tube lens is 200 mm, the magnification of the resulting metalens is 31×. The beam waist $\omega_x$

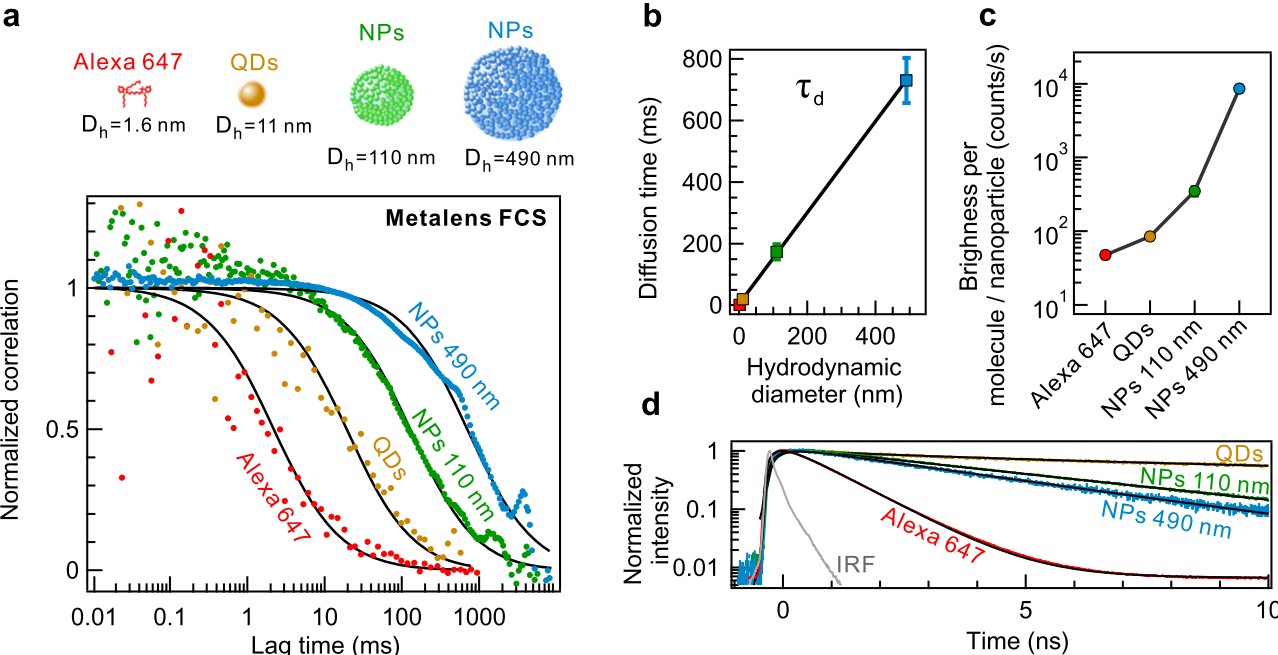

**Fig. 6 | Single nanoparticle detection and size identification. a** Normalized FCS correlation functions of Alexa 647, quantum dots (QDs), plastic nanoparticles (NPs) covered with organic dyes. Corresponding hydrodynamic diameters ($D_h$) are depicted in top panel. Hydrodynamic diameter data of Alexa 647 is adopted from ref. 69., that of QDs from ref. 70, those of NPs from supplier test report. Alexa 647 molecules were excited at 1 mW laser power, whereas QDs and NPs are excited at 100 μW. **b** Retrieved diffusion time ($\tau_d$) of emitters as a function of their $D_h$. Black line corresponds to linear fit following Stokes–Einstein diffusion law. **c** Brightness values of single Alexa 647 molecules and nanoparticles (CRM) retrieved from FCS correlation function analysis. Error bars denote standard deviation of FCS experiment. **d** Fluorescence decays of analyzed emitters. Black lines correspond to fluorescence decay fits. Owing to long lifetime of QDs, repetition rate of laser pulse emission is set to 5 MHz.

found from the focal spot images (Fig. 3) is about 0.9 μm leading to approximately 57 μm spot diameter at the pinhole plane. Therefore, the confocal pinhole size is set to 80 μm.

## Samples of fluorescent molecules and nanoparticles
Alexa Fluor 647 NHS Ester powder (HPLC purity 98%) is purchased from ThermoFisher Scientific (1 mg, catalog number: A20006). Alexa Fluor 647 molecules are dissolved in a phosphate buffer solution from Merck (PBS, pH 7.4, sterile, catalog number: P5244), diluted to 10 μM concentration, and stored at 4 °C in the dark. Nanomolar and picomolar solutions are prepared shortly before the experiment to avoid unwanted nonspecific adhesion to tube walls. High-viscosity experiments are conducted in the presence of glycerol in aqueous solution. Glycerol is obtained from Sigma-Aldrich (spectrophotometric grade >99.5%; catalog number: 191612). Quantum dots with a core/shell structure (CdSe/ZnS) are purchased from ThermoFisher Scientific (Qdot 655 ITK carboxyl quantum dots). Fluorescent nanoparticles with diameters of 110 and 490 nm are purchased from ThermoFischer Scientific (TetraSpeck, Fluorescence Microsphere Sampler Kit). The solutions are placed on a glass slide with a thickness below 0.17 μm to ensure that the working distance of the metalens is sufficient to focus through it.

## Fluorescence time trace data processing
Fluorescence intensity time traces, FCS correlation functions, and TCSPC histograms were processed using SymPhoTime 64 (PicoQuant). The FCS correlation functions retrieved from the metalens measurements are fitted using the 3D-Brownian diffusion model with the background noise contribution[66]:

$$G(\tau) = \frac{(1 - B/F)^2}{N(1 + \frac{\tau}{\tau_d})\sqrt{1 + \frac{\tau}{\kappa^2\tau_d}}} \quad (1)$$

Here, $B$ denotes the background noise (including the uncorrelated out-of-focus fluorescence intensity), $F$ is the total collected fluorescence intensity, $N$ is the number of molecules or nanoparticles, $\tau_d$ is the diffusion time, and $\kappa$ is the aspect ratio of the focal volume. The background noise $B$ is measured by moving the focal volume of the metalens below the fluorescent solution. Based on the correlation amplitude at zero lag time, the number of molecules/nanoparticles is determined as $N = (1 - B/F)^2/G(0)$. The single molecule/nanoparticle brightness is directly deduced from the FCS measurements as $CRM = (F - B)/N$, where $F - B$ denotes the fluorescence intensity collected from the detection volume. The acquisition time for FCS measurements of quantum dots (QDs) and nanoparticles (NPs) is conducted within 200 s, while the Alexa 647 fluorescence has been recorded for 600 s to improve the SNR[67,68]. The apparent detection volume of the metalens is determined as $V = \frac{N_{mol}}{N_A \cdot C}$ with $N_A$ being Avogadro's number, $C$ being the molecule concentration. The confocal effective volume is determined from the point spread function data (Fig. 3) as follows: $V_{eff} = \pi^{3/2}\omega_x\omega_y\omega_z = 56 fl$ with $\omega$ being the corresponding beam waist.

As Alexa 647 molecules tend to exhibit microsecond blinking, the FCS data from control experiments with a high NA objective lens (NA = 1.2) is fitted by the diffusion model with a triplet term as follows:

$$G(\tau) = \frac{1 + \frac{T}{1-T}e^{-\tau/\tau_T}}{N(1 + \frac{\tau}{\tau_d})\sqrt{1 + \frac{\tau}{\kappa^2\tau_d}}} \quad (2)$$

With $T$ being the fraction of molecules in the dark (triplet) state, and $\tau_T$ being the triplet state blinking time.

The fluorescence decays are fitted by exponential functions via an iterative reconvolution that takes into account the instrument response function (IRF). FWHM of the instrument response function (IRF) amounts to 160 ps. Fitting undergoes via the exponential

reconvolution of IRF. Mono-exponential approximation appropriately fits all the decays except for QDs.

## Data availability

All data generated in this study are available in the main text and the supplementary information. Raw data that support the findings of this study will be made available from the corresponding authors upon request.

## Code availability

The codes used in this study are available from the corresponding authors upon request.

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

## Acknowledgements

This work was financially supported by the National Research Foundation (NRF) grants (RS-2023-00266110, NRF-2020R1A5A1019649, NRF-2023M3K5A109482011, NRF-2022M3C1A3081312, NRF-2022M3H4A1A02074314, NRF-2019R1A5A8080290, RS-2023-00283667, RS-2023-00302586) funded by the Ministry of Science and ICT (MSIT) of the Korean government, the POSCO-POSTECH-RIST Convergence Research Center program funded by POSCO, and the POSTECH-Samsung Semiconductor Research Center (IO201215-08187-01) funded by Samsung Electronics. I.K. acknowledges the NRF *Sejong* Science fellowship (NRF-2021R1C1C2004291) funded by the MSIT of the Korean government. Y.K. and D.K.O. acknowledge the Hyundai Motor *Chung Mong-Koo* fellowships. Y.K. acknowledges the NRF Ph.D. fellowship (NRF-2022R1A6A3A13066251) funded by the Ministry of Education of the Korean government. J.J. acknowledges the NRF *Sejong* Science fellowship (RS-2023-00209560) funded by the MSIT of the Korean government.

## Author contributions

I.K., A.B., and J.R. conceived the initial idea, and developed the concept. A.B. and H.P. constructed the time-resolved confocal microscope setup. A.B. performed fluorescence measurements for single molecule and nanoparticle detection. Y.K. and J.J. designed and characterized the dual-wavelength metalenses. D.K.O fabricated the metalens devices. All authors wrote the manuscript, and contributed to the discussion and analysis. I.K. and J.R. guided the entire project.

## Competing interests

The authors declare no competing interests.
