## [Peer Review File · Nature Communications]

Dual-Wavelength Metalens Enables Epi-Fluorescence Detection from Single MoleculesREVIEWER COMMENTS

Reviewer #1 (Remarks to the Author):

The manuscript entitled, "Flat Dual-Wavelength Metalens Enables Epi-Fluorescence Detection from Single Molecules," presents a nanostructured metalens that the authors use for fluorescence correlation spectroscopy (FCS). Although the authors do not demonstrate a complete compact and portable device, the demonstration that a metalens can identify single molecules using FCS is a significant step toward a future compact and portable device. To the best of my knowledge, the authors make a truthful claim that this is the first demonstration of a metalens with sufficient performance for single-molecule FCS, and thus I recommend publication after the authors address the following points:

(1) On line 44, where the authors discuss compact imaging devices with single-particle sensitivity, another relevant citation could be included:

Euan McLeod, T. Umut Dincer, Muhammed Veli, Yavuz N. Ertas, Chau Nguyen, Wei Luo, Alon Greenbaum, Alborz Feizi, and Aydogan Ozcan, "High-throughput and label-free single nanoparticle sizing based on time-resolved on-chip microscopy," *ACS Nano*, 9 (3), 3265-3273 (2015).

(2) On line 49, where the authors discuss broadband metalenses, another relevant citation could be included:

Mikael P. Backlund, Amir Arbabi, Petar N. Petrov, Ehsan Arbabi, Saumya Saurabh, Andrei Faraon, W. E. Moerner, Removing orientation-induced localization biases in single-molecule microscopy using a broadband metasurface mask, *Nature Photonics*, 10, 459–462 (2016),

(3) On lines 139-140, the authors give the metalens conversion efficiencies of 71-76%. Since this is quite far from 100%, doesn't this lead to a large background signal? How do the authors compensate for this background, or does it turn out to be not that significant?

(4) On line 157, the authors give the numerical aperture (NA) of the lens as 0.6. But the focused laser spot size is significantly larger than $\lambda/(2NA)$. The authors should comment on this discrepancy and make it clear in the manuscript that the NA is based on

the physical diameter and focal length of the lens and not its resolving power.

(5) On line 204, the authors state that they cannot resolve a 5 us characteristic time because of the "lack of single molecule brightness." However, earlier it was clear that single molecules could be observed in their system. More specifically, what is meant here by "single molecule brightness"?

(6) On line 215, the authors state that the 280 fl FCS volume exceeds the PSF volume. The authors should give the numerical value of the PSF volume so that it is easy to tell what level of difference there is between the two volumes.

(7) On line 272, there is a typo with the word "metal."

Reviewer #2 (Remarks to the Author):

The authors of the manuscript titled "Flat Dual-Wavelength Metalens Enables Epi-Fluorescence Detection from Single Molecules" design, fabricate and experimentally test a flat metalens that can replace complex and cumbersome objective lenses of high NA to perform single molecule fluorescence spectroscopy.

I think the work is interesting with some good experimental results presented. However, I have some major concerns that I would like the authors to address before considering this manuscript for publication.

Major comments:

1) The authors say in the abstract, and repeat in several places within the manuscript, that "...the metalens enables real-time monitoring of individual fluorescent nanoparticle transitions and identification of hydrodynamic diameters..."

Is the metalens that allows for the real-time monitoring, or does the metalens just simplify and miniaturize the microscope apparatus? Although this is the main novelty that the authors' claim, there are no data or results presented showing the difference between having the metalens or not.

-Does the metalens offer an measurement improvements?
-Does the metalens allow us to reach the single molecule/particle limit, while conventional optics using the same techniques of FCS etc does not?
-The authors also claim in the conclusions/discussion that "Observing temporal single fluctuations provides access to molecular parameters...", which I agree. But how does the metalens does this, and what is the improvement from current techniques?
I am afraid that the authors have not answered this question with the results they presented or during their discussion, which is very hard to judge the novelty of the work.

2) In the conclusions/discussion, the authors claim that "These results confirm that metalenses can compete with the most complex and costly objective lenses with high NA, typically employed in single molecule studies". However, they do not provide any comparative results/evidence to have actually proven this point.
Similarly for the miniaturization claim. Although it is easy to see that the experimental set-up is miniaturized, it would be beneficial to say by how much.

3) The authors measure data from an Alexa647 molecule (diameter=1.6nm), a QD (D=11nm) and two nanoparticles of diameters 110nm and 490nm. Also, during their discussion they keep alternating between using the words "molecule", "particle" or "nanoparticle", which was very confusing. I think they should just use one word, for example "fluorescent object" and clarify early in the manuscript what type of fluorescent objects they measured and why they chose these ones. Also, I didn't quite understand what are the nanoparticles made of? Are they actually fluorescent?

4)The basic principles used to design the metalens have been proposed before: a dielectric meta-atom is used, whose size dimensions change concentrically to induce a different phase to transmitted waves and focus them (or collimate an emission from a dipole source). Here, the design allows to be used for two different wavelengths. Given that there is a lot of literature on very similar metalens designs, I think the authors should either cite them (or at least the first couple papers), explain the basic principles of the design and what is unique here.

5) The authors say that the metalens was "optimized for fluorescence detection in the spectral band between 655 and 700nm...". However, all the results in Fig.3 characterize the metalens at 635nm and 670nm. Also, earlier it is claimed that the metalens focuses the excitation beam that has a wavelength of 635nm. Why is the metalens not optimized for the excitation beam? Is this a typo, or is there something else going on that is not explained in detail.

6) When the authors start discussing their experiments of the metalens FCS of single molecules, they state: "We incorporate an achromatic doublet lens to adjust the beam width to the size of the metalens". Why do they authors need to do this? Why not design a larger metalens for the size of the beam width they have? It will probably give them larger NA than the current 0.6

7) It was not clear to me how the authors determined that they were measuring a single molecule/particle. I'm not too familiar with FCS, but they should not expect the readers of Nature Communications to also be. Can the authors elaborate a bit in the main manuscript how their measurements allow them to measure fluorescence from a single molecule/particle?

Although the idea is good and the work thorough, I am not sure if there is enough novelty to justify publication in Nature Communications. However, I would like the authors to try to address the points above. If it is done in a satisfactory way and indeed show that their metalens significantly improves the measurement accuracy or allows us to measure beyond the current limits, then I will reconsider.

Reviewer #3 (Remarks to the Author):

This article offers an intriguing application of high NA dielectric metalenses towards single molecule sensing and fluorescence correlation spectroscopy. After surveying this work's figures and text, one can perceive the value of this research lies in the ability to create

miniaturized sensing platforms based on thin lenses of varying wavelength sensitivities onto a single substrate and laterally displacing them across the differing laser lines used for this metrology. Further value could be drawn from this methodology if the promise of compact single molecule sensing/integration devices can be realized from this proof of concept due to the high performance afforded by high NA thin lenses.

While the constructed device operates as a proof of concept, my main question about this research paper's value to the community lies in the benchmarking of this technique against standard fluorescence microscopy. The kind of fluorescence microscopy discussed in this work traditionally encompasses a range of very sophisticated instrumentation that can achieve high performance imaging not only using conventional optics, but also using additional optical excitation and collection techniques. For any new method--particularly one that champions an imaging technology that has gained traction in recent years--to make an impression in this community would require the author to clearly specify the metrics that justify the use of this technique against conventional methods. As it stands, I believe the discussion section would be strengthened by a summative statement or table that specifies clearly where metalenses have an advantage over high performance high NA objectives. E.g. lines 246 to 248 in the manuscript provide a compelling argument in favor of using metalenses due to the detectability of nanoparticles below the diffraction limit. If this is indeed an attribute that allows the lens to improve over conventional high performance objectives, this should be made clear in the discussion even if the noise associated with the structure hinders the collection of clear data. Otherwise, the metalens appears to merely act as a substitute (a chromatically limited one too) in this experiment's image relay system.

The paper already describes some of the deficiencies associated with metalenses and their realization in practice. First, replicating the designed phase functions is made difficult by the precision required during nanofabrication, leading to some of the efficiency losses mentioned in the discussion section of the work. Second, their limited chromatic bandwidth inevitably necessitates the use of different lenses when operating along the other wavelengths in a fluorescence microscope's laser line--something obviated in a traditional microscope objective. This behavior is already well known in a community accustomed to the deficiencies of single surface lenses based on highly chromatic pillar designs. What

should be highlighted more in the metalens-centered figures (e.g. Fig 2 and 3) is a clear correlation of how the theoretical and fabricated lenses' focusing properties benefit this experiment. Was there a better fabricated metalens with a higher focusing efficiency that can achieve detection better? Do different lens NAs provide advantages in their PSF and the ability to detect the particles under investigation? Are there other libraries that can function better than the current amorphous silicon fins?

I do appreciate the work that went into capture the actual lens' PSF--the FWHM demonstrates the feasibility of applying such a structures towards detecting some of the smaller structures investigated in this study (like the Alexa dye). However, I wish that Figures 2 and 3 were better substantiated with information like this that really rationalizes the power of using these high NA flat optics.

I also appreciate the volume of data collection that went into completing Figures 4 and 5. While it suggests flaws in the nature of the hardware when compared against standard equipment (especially due to the flaws of the fabricated nanostructures), it validates the use case depicted by this paper and shows the feasibility of this application should the fundamental issues of metalens fabrication and performance be resolved in future designs.

I'd like to request the author to include a stronger summative statement with a table that enumerates the strengths of the metasurface based imaging system described in this paper against the conventional approach. It is inevitably going to suffer in performance when compared against traditional fluorescence microscopy, but these metrics need to be clearly laid out so that the researchers who follow up on this line of work can clearly reference the benchmark demonstrated here.

A softer request is to substantiate Figures 2 and 3 with metalens performance metrics that more directly correlate to the function of the experiment at hand. Phase functions are phase functions--how well you meet them should be reflected by the efficiency and ought to be more of a supporting information kind of inclusion. Is there data that you already have in the SI that can better reflect your unique metalens' designs benefits for FCS and single molecule detection?

These small changes aside, I believe this paper has value to add to the literature for the novel photonics and molecule sensing/device space.

Reviewer #1 (Remarks to the Author):

The manuscript entitled, "Flat Dual-Wavelength Metalens Enables Epi-Fluorescence Detection from Single Molecules," presents a nanostructured metalens that the authors use for fluorescence correlation spectroscopy (FCS). Although the authors do not demonstrate a complete compact and portable device, the demonstration that a metalens can identify single molecules using FCS is a significant step toward a future compact and portable device. To the best of my knowledge, the authors make a truthful claim that this is the first demonstration of a metalens with sufficient performance for single-molecule FCS, and thus I recommend publication after the authors address the following points:

Our response:

We thank the reviewer for the appreciation of our work. We will address the points raised by the reviewer in the responses below.

(1) On line 44, where the authors discuss compact imaging devices with single-particle sensitivity, another relevant citation could be included:

Euan McLeod, T. Umut Dincer, Muhammed Veli, Yavuz N. Ertas, Chau Nguyen, Wei Luo, Alon Greenbaum, Alborz Feizi, and Aydogan Ozcan, "High-throughput and label-free single nanoparticle sizing based on time-resolved on-chip microscopy," ACS Nano, 9 (3), 3265-3273 (2015).

Our response:

We added this relevant reference in the Introduction and the following statement summarizing it (Page 3):

"Alternatively, hand-held lensfree holographic imaging platforms could provide single nanoparticle observation above 40 nm.¹⁸"

(2) On line 49, where the authors discuss broadband metalenses, another relevant citation could be included:

Mikael P. Backlund, Amir Arbabi, Petar N. Petrov, Ehsan Arbabi, Saumya Saurabh, Andrei Faraon, W. E. Moerner, Removing orientation-induced localization biases in single-molecule microscopy using a broadband metasurface mask, Nature Photonics, 10, 459–462 (2016),

Our response:

We thank the reviewer for providing another relevant reference for highlighting metasurface technology importance for single molecule community. We added the reference and following statement that summarizes it (Page 3):

"Dielectric metasurfaces operating as phase and polarization masks in the microscope Fourier plane are able to get rid of localization biases in single-molecule imaging.³¹"

(3) On lines 139-140, the authors give the metalens conversion efficiencies of 71-76%. Since this is quite far from 100%, doesn't this lead to a large background signal? How do the authors compensate for this background, or does it turn out to be not that significant?

Our response:

We thank the reviewer for his/her comment. Indeed, we observe a non-negligible background intensity, which is likely to be caused by incomplete phase conversion. This background fluorescence light from out-of-focus molecules limits the achievable range of correlation amplitudes as we state in the main text: "The collected non-diffracted fluorescence light which is emitted away from the focal volume affects the correlation amplitude at high concentrations. In contrast, the residual background noise of the metalens structure restrains the correlation amplitude to below approximately 15×10^{-3} at low concentrations." Nevertheless, in order to fit the FCS data and extract molecular parameters we utilize a conventional 3D-Brownian diffusion model with the non-zero background light (Equation 1): $G(\tau) = \frac{(1-B/F)^2}{N(1+\frac{\tau}{\tau_d})\sqrt{1+\frac{\tau}{\kappa^2\tau_d}}}$, where B is the

background noise, F is the total collected fluorescence intensity in the condition when the metalens focuses light inside the fluorescent solution. The background light is measured by focusing the metalens significantly below the interface between a glass slide and fluorescence solution. In the case of nanomolar concentrations of fluorescent molecules/nanoparticles the background intensity varies between 10 % and 20 % of total collected fluorescence intensity.

(4) On line 157, the authors give the numerical aperture (NA) of the lens as 0.6. But the focused laser spot size is significantly larger than $\lambda/(2NA)$. The authors should comment on this discrepancy and make it clear in the manuscript that the NA is based on the physical diameter and focal length of the lens and not its resolving power.

Our response:

The observation that the metalens, despite having a specified numerical aperture (NA) of 0.6, exhibits a focused laser spot size significantly larger than $\lambda/2NA$, warrants a comprehensive investigation. Several factors may contribute to this discrepancy, with the primary reasons being phase mismatch and manufacturing imperfections, both of which significantly contribute to the enlargement of the focal spot.

The phase mismatch arises due to geometric limitations encountered during the fabrication of nanostructures when accounting for the material's n and k values. To elaborate, our ability to create nanostructures using electron beam lithography (EBL) is bound by the constraints imposed by amorphous silicon (a-Si:H), thereby determining the achievable dimensions in terms of length, width, and height. Consequently, practical constraints on the attainable shapes of nanostructures inevitably lead to a disparity between the theoretical phase map and the actual matched phase map. As illustrated in the figure below, a conspicuous disparity exists when comparing the theoretical phase difference map with the post-matching phase difference map.

Figure S9. (a) Phase difference between 635 nm and 670 nm phase maps in theoretical case. (b) Phase difference between 635 nm and 670 nm phase maps in matched case.

Manufacturing imperfections come from variations in geometry between the target nanostructure and the fabricated counterpart. As depicted in the figure below, while our intended structure is a perfect cuboid, the fabricated structure appears more rounded, with oblique sides deviating from the ideal 90-degree angle due to imperfect etching conditions. This discrepancy induces unintentional phase modulation, significantly influencing the broader full-width-half-maximum observed in our metalens.

Figure S10. (a) The oblique and top view of target nanostructure. (b) The oblique and top view of fabricated nanostructure.

Additionally, edge effects, stemming from non-ideal behaviors near the lens periphery, and grating effects, especially in nanoscale structures, may affect the focal properties. We have added the investigation of reasons for enlargement of the focal spot in Supplementary Information (section S9).

As the reviewer mentioned, the experimental focal spot is larger than theoretically designed focal size, but the focal length is positioned well. Therefore, we have defined the effective NA, which is determined by physical diameter and focal length, and explained the reasons for larger focal spot in our manuscript (Page 11) as below.

“The experimental focal spots are enlarged because of several reasons including phase mismatch and manufacturing imperfection (see details in Supplementary Information section S9). Therefore, the experimentally measured NA is the effective NA, which is determined by physical diameter and focal length. The images confirm that the focal points are positioned at a distance of 330 μm corresponding to effective NA = 0.6, which is large enough to focus laser light through a thin glass slide.”

(5) On line 204, the authors state that they cannot resolve a 5 us characteristic time because of the "lack of single molecule brightness." However, earlier it was clear that single molecules could be observed in their system. More specifically, what is meant here by "single molecule brightness"?

Our response:

We thank the reviewer for his/her comment. We acknowledge that this statement may cause confusion for the reader. We achieve fluorescence brightness per single Alexa647 molecule of around 50 counts/s (expressed in a photon rate at our detector) using the metalens platform. This is retrieved from the correlation functions of diffusing molecules in the millisecond timescale, which we were able to clearly capture. Based on the correlation function amplitudes we directly extract the value of number of molecules (N_{mol}) present in the detection volume, while the fluorescence photon rate after background subtraction ($F-B$) is the measure of total detected fluorescence intensity from the molecules. Dividing $F-B$ by N_{mol} yields the single molecule brightness, which we also designate by CRM throughout the manuscript. However, the FCS correlation function exhibits nearly 2 orders of magnitude higher noise in the microsecond time range which originates from the FCS signal-to-noise ratio (SNR) definition at short lag times:

$$SNR = CRM \cdot \sqrt{T \cdot \delta\tau} / \sqrt{1 + 1/N_{mol}},$$
 where T is the total signal acquisition time, $\delta\tau$ is the correlator

channel width (microsecond times in this case). Therefore, resolving 5 μ s blinking term with the same quality as diffusion term would require to increase the acquisition time T by 2 to 3 orders of magnitude to compensate the reduction of $\delta\tau$. This is hardly possible in a practical experiment. We added an extra reference to Methods in the main text (Page 14):

“We directly retrieve the single molecule brightness by dividing the fluorescence intensity collected from the detection volume by the number of molecules (Methods).”

(6) On line 215, the authors state that the 280 fl FCS volume exceeds the PSF volume. The authors should give the numerical value of the PSF volume so that it is easy to tell what level of difference there is between the two volumes.

Our response:

We thank the reviewer for his/her comment. We acknowledge that we should more specifically indicate the difference between the FCS effective volume and the PSF volume. The PSF volume for the metalens amounts to 56 fl (more details in Methods) based on the Gaussian focal volume equation: $V_{eff} = \pi^{3/2} \omega_x \omega_y \omega_z$, where $\omega_{x/y/z}$ are focal beam waists along corresponding axes. This difference is around 5 times which is rather significant. In fact, the PSF volume reflects only the focusing volume of the excitation volume. However, the FCS effective volume reflects the convolution of the excitation focal volume and the detection volumes at wavelengths within the fluorescence band. Therefore, the PSF and effective volumes are equal only under ideal excitation conditions (no astigmatism or coma), ideally achromatic collecting lens, and perfectly fit pinhole size. As these conditions are rarely feasible even in the FCS experiments with the best conventional optics, these values appear different in our case owing to remaining chromatic aberrations of the metalens within the Alexa 647 fluorescence band and aberrations induced by the presence of the confocal glass slide. Lastly, we would like to point out here, that sufficiently strong chromatic aberrations do not even enable reaching single molecule FCS detectability, as

we show now with conventional aspheric lenses that provide sharp single wavelength focusing. That signifies, that the single molecule sensitivity can be unreachable due to extreme mismatch in overlap of excitation and detection volumes even for rather high NA lenses, *i.e.* expanding the confocal effective volume to quasi-infinite values. We added the factor of difference between two aforementioned volumes and a comment about chromatic aspheric lenses in the main text (Page 14).

“The apparent FCS detection volume in the aqueous solution amounts to 280 fl which exceeds the confocal effective volume determined from the PSF data by around 5 times (see details in Methods).”

“Moreover, remaining chromatic distortion can contribute to the effective volume expansion, as it has been clearly demonstrated on near-unity NA aspheric lenses.⁵²”

(7) On line 272, there is a typo with the word "metal."

Our response:

We apologize for the typo. The typo has been corrected.

Reviewer #2 (Remarks to the Author):

The authors of the manuscript titled "Flat Dual-Wavelength Metalens Enables Epi-Fluorescence Detection from Single Molecules" design, fabricate and experimentally test a flat metalens that can replace complex and cumbersome objective lenses of high NA to perform single molecule fluorescence spectroscopy.

I think the work is interesting with some good experimental results presented. However, I have some major concerns that I would like the authors to address before considering this manuscript for publication.

Our response:

We thank the reviewer for thoroughly evaluating our work. We will address the raised questions and concerns in the point-by-point responses below.

Major comments:

1) The authors say in the abstract, and repeat in several places within the manuscript, that "...the metalens enables real-time monitoring of individual fluorescent nanoparticle transitions and identification of hydrodynamic diameters..."

Is the metalens that allows for the real-time monitoring, or does the metalens just simplify and miniaturize the microscope apparatus? Although this is the main novelty that the authors' claim, there are no data or results presented showing the difference between having the metalens or not.

-Does the metalens offer any measurement improvements?

-Does the metalens allow us to reach the single molecule/particle limit, while conventional optics using the same techniques of FCS etc does not?

-The authors also claim in the conclusions/discussion that "Observing temporal single fluctuations provides access to molecular parameters...", which I agree. But how does the metalens do this, and what is the improvement from current techniques?

I am afraid that the authors have not answered this question with the results they presented or during their discussion, which is very hard to judge the novelty of the work.

Our response:

We acknowledge the reviewer's concerns. Indeed, the metalens is used here as part of a miniaturization platform for single molecule detection. One can reach a single molecule by using objective lenses of the most sophisticated design, high cost and NA that are deprived both from spherical and chromatic aberrations. Nevertheless, the low and moderate NA objective lenses do not allow single molecule detection or reach only borderline sensitivity owing to rather inefficient molecule excitation and extremely limited collection efficiency of fluorescence photons. To provide an adequate comparison between our metalens platform performance and the conventional optics, we record FCS data of diffusing Alexa 647 molecules by a set of lenses of different NA and chromatic aberration presences, as these two parameters are the most crucial for single molecule sensitivity. The fluorescence is excited and collected by two aspheric lenses of NA = 0.18 and 0.54 and objective lenses of NA between 0.15 and 1.2. We show that the aspheric lens of similar NA that also has large volume and cost doesn't allow reaching single molecule sensitivity under

same excitation/detection conditions as the metalens. Moreover, our metalens exceeds the performance of objective lenses with $NA < 0.5$ by all critical FCS parameters such as single molecule brightness, effective FCS volume, and diffusion time of molecules. Then, the metalens-enabled single molecule brightness and diffusion time fall short of those retrieved by $NA = 0.5$ objective lens, the effective volume is still close to the conventional objective lens of a similar NA. We admit that the objective lenses of NA above 1 feature excellent single molecule sensitivity as they provide subwavelength focusing and high photon collection efficiency together with broadband chromaticity. Even though, those objective lenses have been optimized by industries through decades and are utilized by the single molecule community, those objective lenses remain rather costly and are not compatible with portable on-chip sensing devices. As for our metalens performance, the limited single molecule brightness is attributed mainly to the non-ideal conversion efficiency, focusing efficiency and not broadband achromaticity as all these factors play a pivotal role in the resulting collection efficiency. As the forward-design metalens functionalities such as NA, focusing efficiency, achromaticity, size cannot be boosted together to the same values as state-of-the-art conventional refractive optics due to theoretical limitations (refs.32,33). As we discuss in this work, the single molecule detectability requires a trade-off of all those functionalities for experimental realization.

Most importantly, we would like to point out that the novelty of our work is mainly represented by the first demonstration of ability of a metalens system to detect single molecule that is hardly achievable even with conventional low NA objective lenses. Moreover, future potential integration of the metalens with on-chip sensing devices could relax the requirements for the metalens diameter (as in our case we operate under conventional confocal illumination) and could enable more degrees of freedom to boost the performance to higher levels. In the newly added data, we additionally show the miniaturization degree of the microscope apparatus as the main advantage of the metalens and the quantitative comparison with sizes of conventional optics. We included the comments and Figure 5 in the main text (Pages 15 and 16) and Supplementary Information (Figure S15).

Figure 5. Metalens performance comparison against conventional optics. (a) Set of lenses employed for comparative study with indicated size dimensions: metalens, two single-element aspheric lenses of NA = 0.18 and NA = 0.54, and achromatic objective lenses of NA between 0.15 and 1.2. (b) Alexa 647 single molecule brightness detected by metalens and aspheric lenses. As FCS correlation is not detectable by aspheric lenses, single molecule brightness is considered to equal zero. (c) Alexa 647 single molecule brightness detected by metalens and achromatic objective lenses. (d) Alexa 647 molecule diffusion time detected by metalens and achromatic objective lenses. (e) Confocal effective volume observed by metalens and achromatic objective lenses.

Figure S15. (a), (b) FCS correlation functions of diffusing Alexa647 molecules acquired using aspheric lenses of NA = 0.18 and NA = 0.54, respectively. (c), (d), (e), (f), (g), (h), (i) FCS correlation functions of diffusing Alexa647 molecules acquired using achromatic objective lenses of NA = 0.15, NA = 0.25, NA = 0.3, NA = 0.5, NA = 0.8, NA = 0.95, NA = 1.25, respectively. The concentration of Alexa 647 is fixed at 6 nM in these measurements.

“In order to provide an adequate comparison between our metalens performance and the conventional optics, we record FCS data of diffusing Alexa 647 molecules by a set of lenses of

different NA and chromatic aberration presences (Figure 5a). The fluorescence from Alexa 647 solution is excited and collected by two aspheric lenses of NA = 0.18 and 0.54 and objective lenses of various NA in the range of 0.15 and 1.2 (lens sizes shown in Table S4). The corresponding raw FCS correlation traces are shown in Figure S15. We observe that the aspheric lenses of NA close to that of the metalens or below do not enable reaching single molecule sensitivity under identical excitation/detection conditions (Figure 5b). The lack of sensitivity comes from the mismatch in overlap between the excitation and detection volume of the single-element aspheric lens due to its intrinsic chromatic distortion.^{51,52} Employing achromatic multi-element objective lenses clearly boosts the single molecule fluorescence intensity (Figure 5c), although it still remains undetectable for objective lenses with NA < 0.25. The designed metalens exceeds the performance of objective lenses with NA < 0.5 by all critical FCS parameters such as single molecule brightness, effective FCS volume, and diffusion time of molecules (Figures 5c-e). Although the metalens-enabled single molecule brightness and diffusion time fall short of those retrieved by NA = 0.5 objective lens, the effective volume comes close to the conventional objective lens trend.”

2) In the conclusions/discussion, the authors claim that "These results confirm that metalenses can compete with the most complex and costly objective lenses with high NA, typically employed in single molecule studies". However, they do not provide any comparative results/evidence to have actually proven this point.

Similarly for the miniaturization claim. Although it is easy to see that the experimental set-up is miniaturized, it would be beneficial to say by how much.

Our response:

We acknowledge the importance of reviewer’s comment. As FCS studies are typically conducted with high NA objective of good quality of aberration correction, we included the statement quoted by the reviewer. As stated in the previous response, we have added a study comparing metalens performance for single molecule detection, chromatic aspheric lenses of lower and similar NA, and achromatic objective lenses with a wide range of NA. The metalens performs experimentally better than objective lenses of NA ≤ 0.3 and aspheric lenses of the same NA. Taking into account simulated design focusing efficiency of the metalens at the excitation and emission wavelengths of the fluorophore, metalens should compete with the objective lenses of similar NA, e.g. the objective lens with NA = 0.5 shown in newly added Figure 5. We admit that the most complex objective lenses with NA ~ 1 or above are better performing, therefore, we change the statement from “...metalenses can compete with the most complex and costly objective lenses with high NA, typically employed for single molecule studies” to “...the metalens platform can compete with complex and costly objective lenses for single molecule sensing and dynamics studies”. We also provide the sizes of the employed lenses in a table to highlight the setup miniaturization owing to metalens platform (in Supplementary Information section S15).

Table S4. Metalens size comparison against conventional optics.

Lens type	Numerical aperture	Height (mm)	Width (mm)
Metalens	0.6	0.5·10 ⁻³ (structure) 0.5 (substrate)	0.5

Aspheric lenses	0.18	3	7.5
	0.54	9	25
Achromatic objective lenses	0.15	30	25
	0.25	40	25
	0.3	40	25
	0.5	50	25
	0.8	50	25
	0.95	65	33
	1.2	65	34.5

3) The authors measure data from an Alexa647 molecule (diameter=1.6nm), a QD (D=11nm) and two nanoparticles of diameters 110nm and 490nm. Also, during their discussion they keep alternating between using the words "molecule", "particle" or "nanoparticle", which was very confusing. I think they should just use one word, for example "fluorescent object" and clarify early in the manuscript what type of fluorescent objects they measured and why they chose these ones. Also, I didn't quite understand what are the nanoparticles made of? Are they actually fluorescent?

Our response:

We acknowledge reviewer's concern about the confusion caused by using similar terms. We have changed words "particle" to "nanoparticle" in the main text and Figure 6c. Although, we admit a term "fluorescent object" may generalize the studied objects, we kindly disagree with the reviewer's suggestion to use it as a unified term. We believe that using this would lead to confusion between the emitters of rather different size, brightness, and photoluminescence physics and, more importantly, undermine the main achievement of this work which is the metalens-enabled fluorescence detection from single organic molecules in the absence of objective lens. We selected an Alexa 647 molecule as the main studied fluorescent object, since it is a benchmark fluorophore for biology including bioimaging and biosensing, and it yields highly photostable conjugates and optimized blinking behavior. Also, Alexa 647 has emerged as a benchmark fluorophore in conventional FCS studies in aqueous solutions and living cells. Additionally, we employ the nanoparticles that yield emission spectra that partly overlap with that of Alexa 647 and feature different sizes. Even though all the objects have emitters of various chemical nature with slightly different fluorescence spectra they all can be detected by the metalens microscope apparatus. By collecting the data from all those objects, we confirm that the FCS metalens platform is indeed sensitive to fluorescence object size, concentration, and the solution viscosity. Responding to the last reviewer's question in the comment, the fluorescent nanoparticles are plastic nanoparticles coated by organic dyes of various colors including far-red dyes with fluorescence band emitting around 680 nm, close to the emission maximum of Alexa647 (670 nm). Those fluorescent nanoparticles are commercially available (ThermoFischer Scientific, "TetraSpeck, Fluorescence Microsphere Sampler Kit") that are utilized as resolution targets of single- and multi-color fluorescence imaging. CdSe/ZnS quantum dots exhibit emission maximum around 655 nm.

We added corresponding comments to the main text (Pages 4 and 5).

“We employ Alexa Fluor 647 as a benchmark fluorophore commonly used in bioimaging and biosensing techniques, and it yields highly photostable conjugates and optimized blinking behavior under red light excitation.⁴¹⁻⁴³ Additionally, Alexa 647 has emerged as a standard fluorophore in conventional FCS studies in aqueous solutions and living cells.^{44,45}”

“The nanoparticles yield emission spectra that partly overlap with that of Alexa 647 and, hence, appear compatible with the metalens platform.”

4)The basic principles used to design the metalens have been proposed before: a dielectric meta-atom is used, whose size dimensions change concentrically to induce a different phase to transmitted waves and focus them (or collimate an emission from a dipole source). Here, the design allows to be used for two different wavelengths. Given that there is a lot of literature on very similar metalens designs, I think the authors should either cite them (or at least the first couple papers), explain the basic principles of the design and what is unique here.

Our response:

We thank the reviewer for this remark. In order to employ the metalens for single molecule sensing under confocal illumination, the metalens have to achieve a certain trade-off of properties as below.

1. Working distance (focal length) > 200 μm (comfortable distance to focus through confocal glass slide)
2. High NA >= 0.6 which resulted in diameter ~ 500 μm
3. Working wavelength can cover the excitation and collection wavelengths of fluorescent beads. (635 nm & 670 nm)
4. Average transmittance > 85 %
5. Polarization-insensitivity: metalens can working in all polarization condition

To achieve these goals, the co-polarization term is selected as the phase modulation method.

Figure S3. Principle of propagation phase.

The figure below shows the conversion efficiencies by using co-polarization term is much larger than the efficiencies by using cross-polarization term.

Figure S7. (a) Conversion efficiency at 635-nm incidence by using co-polarization term. (b) Conversion efficiency at 670-nm incidence by using co-polarization term. (c) Conversion efficiency at 635-nm incidence by using cross-polarization term. (d) Conversion efficiency at 670-nm incidence by using cross-polarization term. The average conversion efficiency at 635-nm of (c) is 0.6582, and that of (d) is 0.6221.

We meticulously determined the dimensions of length and width, pixel by pixel, and matched precisely with the theoretical phase map, minimizing phase differences. In order to provide a lucid rationale for our choice of this design methodology, we included a comprehensive comparison of conversion efficiencies. This explanation, along with further details, can be found in both the main manuscript (Page 8 and 9) and the supplementary information provided.

“This result represents the conversion efficiency obtained when using the co-polarization term, which is more than 5% higher than the conversion efficiency observed when using cross-polarization (Figure S7).”

As the reviewer mentioned in the comment, we have summarized the related research. We also would like to point out that 6 references related to achromatic metalens design were prior included in the text (refs. 25-30). We summarize the reference papers on the multi-wavelength and polarization-insensitive metalenses in a table. We add this table in the Supplementary Information (section S11) as below.

Table S1. Chromatic aberration correction, size, NA, and focusing efficiency of the metalenses presented in this work and prior arts.

Operating wavelengths (nm)	Feature	Size (μm)	Effective NA	Focusing efficiency (%)	Reference

635 / 670	Single layer	500	0.6	14.1 / 37.1	This work
470 to 670	Achromatic; single layer	220	0.02	~20	1
465 / 548 / 600 / 620	Inverse design, single layer	115	0.3	42.8 / 42.4 / 38 / 33.3	2
1180 / 1400 / 1680	Vertically stacked multi-layer	120	0.29	34.5 / 30.7 / 51.1	3
488 / 532 / 633	Double layer	1000	0.55 / 0.55 / 0.58	12.81 / 13.30 / 42.05	4
1200 to 1650 nm	Achromatic; single layer	100	0.24	~ 35	5

In the reference papers, it is noted that the desired properties with a single layer metalens have not been achieved before, and success in meeting the specifications has been reported with metalenses consisting of two or more layers. However, creating metalenses with more than two layers is a challenging and intricate process, often accompanied by alignment issues between the layers. This paper's advantage lies in the research on single-layer metalenses, successfully achieving tight focusing. Furthermore, it demonstrates applicability for single molecule sensing as an extremely valuable functionality for molecular biophysics and biosensing assays. We also added a corresponding note in the main text (Pages 11, 12).

“Nevertheless, the desired properties have not been achieved with single layer metalenses before, as we conduct the metalens performance comparison to achromatic lenses from prior arts (Table S1). As creating multilayer metalenses with several layers is typically accompanied by layer alignment issues, the advantage of the metalens design of this work is implementation of single layer metalens with tight focusing and sufficiently high focusing efficiency at the fluorophore emission wavelength.”

5)The authors say that the metalens was "optimized for fluorescence detection in the spectral band between 655 and 700nm...". However, all the results in Fig.3 characterize the metalens at

635nm and 670nm. Also, earlier it is claimed that the metalens focuses the excitation beam that has a wavelength of 635nm. Why is the metalens not optimized for the excitation beam? Is this a typo, or is there something else going on that is not explained in detail.

Our response:

We apologize for the apparent confusion of the rationale behind choosing design wavelengths. The quoted text by the reviewer "optimized for fluorescence detection in the spectral band between 655 and 700nm..." in the text is attributed to the spectral filter set in our microscope apparatus and the photon detector efficiency. The metalens is designed for the excitation beam 635 nm and Alexa 647 emission maximum at 670 nm. The achromatic focusing at those two wavelengths was experimentally demonstrated in Figure 3. We added a comment in the main text (Page 6).

"The spectral filter set is optimized for fluorescence detection in the spectral band between 655 and 700 nm (see details in Methods)."

6) When the authors start discussing their experiments of the metalens FCS of single molecules, they state: "We incorporate an achromatic doublet lens to adjust the beam width to the size of the metalens". Why do they authors need to do this? Why not design a larger metalens for the size of the beam width they have? It will probably give them larger NA than the current 0.6

Our response:

As the reviewer mentioned in comment, it appears appropriate to design larger metalens for a microscope apparatus that would yield even higher NA for compact imaging system. Indeed, our prior efforts were directed towards the fabrication of large-sized metalenses of diameter ≥ 1 mm; however, we encountered challenges during the manufacturing process. Firstly, the maximum size of single EBL process (EBL model: Elionix 7800) with high-resolution and no edge effect is approximately 500 μm in diameter. Therefore, we divided the metalens into quarters for lithography. Consequently, we aspired to fabricate metalens with 1.1 μm diameter (about 2 times larger than maximum size) necessitating the sequential fabrication of individual lens segments. Unfortunately, alignment difficulties (or stitching errors) arose during the process, resulting in the formation of disjointed metalens segments as illustrated below.

Rvs Figure 1. The metalens with 1.1 μm diameter (a) SEM image, (b) OM image, and (c) PSF result.

This disjointed issue led to decreased metalens efficiency and an increase in error values, consequently impacting the Point Spread Function (PSF) results and molecule detection efficiency (MDE) adversely. That metalens platform also could not reach single molecule sensitivity for FCS measurements. Future advancements in metalens fabrication machinery or processes may offer the possibility of creating even larger metalenses. Apart from technical limitations, there were reported theoretical limits, especially for single layer metalenses, that do not allow to increase the size of metalens and NA without compromising the focusing efficiency as reported in multiple works including refs. 29, 32, 33; while the latter is clearly a crucial parameter for the single molecule sensing. Thus, in the process of selecting the largest size metalens among those fabricated and experimented upon, one with a diameter of 500 μm was ultimately chosen due to its favorable performance for single molecule FCS owing to higher focusing efficiency and sharper focusing. Lastly, the integration of the doublet lens enables adjusting the magnification of the metalens microscope apparatus to the confocal pinhole sizes at our disposal.

7) It was not clear to me how the authors determined that they were measuring a single molecule/particle. I'm not too familiar with FCS, but they should not expect the readers of Nature Communications to also be. Can the authors elaborate a bit in the main manuscript how their measurements allow them to measure fluorescence from a single molecule/particle?

Our response:

First of all, we would like to point out that the FCS intrinsically is a single molecule technique as the correlation arises from time fluctuations of signal owing to single molecule diffusion events through the confocal effective volume. In the case, of the absence of single molecule sensitivity the dynamic fluctuations of fluorescence due to single molecule diffusion will not be detectable and the correlation will oscillate symmetrically around zero, as we showed before for the measurements in the absence of metalens or with the metalens focusing light below the fluorescent solution/glass interface (Figure S13).

Figure S13. (a) FCS correlation function of diffusing Alexa 647 molecules when metalens is out of focus. (b) FCS correlation function of diffusing Alexa 647 molecules when metalens is removed from beam path. Signal acquisition time is identical to FCS measurements shown in Figure 4.

Second of all, the amplitude of the correlation function, total fluorescence intensity, and background intensity (signal measured when the metalens is defocused from the solution) allow us to retrieve the number of molecules steadily present in the metalens detection volume according to Equation 1 (main text). Thus, by reducing the concentration of fluorescent molecules/nanoparticles to sufficiently low level we confirmed that fluorescence signal is perceivable with only one molecule in the detection volume. Moreover, as nanoparticles yield clearly higher brightness than individual molecules, we monitored the fluorescence intensity bursts of individual nanoparticle passes in real-time. In those experiments, the concentrations were reduced below 1 nanoparticle per detection volume to observe repetitive individual nanoparticle transits through the detection volume. As the time went on, we could monitor the signal bursts above the background level (Figure S16c and S17) originating from NPs 490 nm, NPs 110 nm, and QDs 11 nm. We demonstrate those figures below for clarity.

Rvs Figure 2. Fluorescence intensity time traces of NPs 490 nm at stock concentration.

Figure S17. Real-time monitoring of NPs 110 nm transits at (a) 200× dilution ~ 1 nanoparticle in detection volume, (b) 1000× dilution ~ 0.2 nanoparticles in detection volume, (c) background fluorescence measured with pure PBS buffer. Excitation power for measurements in (a–c) is set to 100 μ W. Real-time monitoring of QDs transits at (d) 8 pM ~ 1.3 nanoparticles in detection volume, (e) 1 pM ~ 0.17 nanoparticle in detection volume, (f) background fluorescence measured with pure PBS buffer. Excitation power for measurements in (d–f) is set to 0.25 mW. Owing to low signal, QDs were diluted in glycerol 60 vv% to delay transit time. Bin sizes employed in (a–f) equal 200 ms roughly corresponding to diffusion time of nanoparticles. Horizontal dashed line designates selected threshold which is not exceeded by background noise. (g) Histograms of time traces (a–c) and number of data points above the selected threshold line at two nanoparticle concentrations. (h) Histograms of time traces (d–f) and number of data points above selected threshold line at two QD concentrations.

We enrich the current methodology discussion in the text regarding the extraction of number of molecules, other relevant FCS parameters, and the concentration at which fluorescence is observed from an individual molecule as below (Pages 13 and 14).

“We fit the FCS correlation functions by the 3D-Brownian diffusion model (see Methods) with the fixed beam aspect ratio $\kappa = 10$ and extract the relevant parameters such as molecule diffusion time, number of molecules in the confocal effective volume of the metalens, single molecule brightness (CRM).”

“To correctly identify the number of molecules from the FCS correlation function fits, we account for the background intensity, *i.e.*, the signal acquired when the metalens focuses the laser light below the fluorescent solution reservoir (see Methods). The time traces and FCS correlation functions can be recorded for only 1.7 molecules (Figure 4c) at 10 pM concentrations, consolidating the ultimate sensitivity of the designed metalens. The number of detected molecules scales linearly with the concentration (C) from one to several thousand molecules.”

We also would like to kindly refer the reviewer to our discussion on FCS correlation analysis in Methods section as below.

“Fluorescence intensity time traces, FCS correlation functions, and TCSPC histograms were processed using SymPhoTime 64 (PicoQuant). The FCS correlation functions retrieved from the metalens measurements are fitted using the 3D-Brownian diffusion model with the background noise contribution⁶⁷

$$G(\tau) = \frac{(1 - B/F)^2}{N(1 + \frac{\tau}{\tau_d}) \sqrt{1 + \frac{\tau}{\kappa^2 \tau_d}}} \quad (1)$$

Here, B denotes the background noise (including the uncorrelated out-of-focus fluorescence intensity), F is the total collected fluorescence intensity, N is the number of molecules or nanoparticles, τ_d is the diffusion time, and κ is the aspect ratio of the focal volume. The background noise B is measured by moving the focal volume of the metalens below the fluorescent solution. Based on the correlation amplitude at zero lag time, the number of molecules/nanoparticles is determined as $N = (1 - B/F)^2 / G(0)$. The single molecule/nanoparticle brightness is directly deduced from the FCS measurements as $CRM = (F - B)/N$, where $F - B$ denotes the fluorescence intensity collected from the detection volume. The acquisition time for FCS measurements of quantum dots (QDs) and nanoparticles (NPs) is conducted within 200 s, while the Alexa 647 fluorescence has been recorded for 600 s to improve the SNR.^{68,69} The apparent detection volume of the metalens is determined as $V = \frac{N_{mol}}{N_A \cdot C}$ with N_A being Avogadro's number, C being the molecule concentration.”

Although the idea is good and the work thorough, I am not sure if there is enough novelty to justify publication in Nature Communications. However, I would like the authors to try to address the points above. If it is done in a satisfactory way and indeed show that their metalens significantly improves the measurement accuracy or allows us to measure beyond the current limits, then I will reconsider.

Our response:

We acknowledge the general opinion of the reviewer. We are full of hope that the aforementioned point-by-point responses persuaded the reviewer toward a favorable decision.

Reviewer #3 (Remarks to the Author):

This article offers an intriguing application of high NA dielectric metalenses towards single molecule sensing and fluorescence correlation spectroscopy. After surveying this work's figures and text, one can perceive the value of this research lies in the ability to create miniaturized sensing platforms based on thin lenses of varying wavelength sensitivities onto a single substrate and laterally displacing them across the differing laser lines used for this metrology. Further value could be drawn from this methodology if the promise of compact single molecule sensing/integration devices can be realized from this proof of concept due to the high performance afforded by high NA thin lenses.

While the constructed device operates as a proof of concept, my main question about this research paper's value to the community lies in the benchmarking of this technique against standard fluorescence microscopy. The kind of fluorescence microscopy discussed in this work traditionally encompasses a range of very sophisticated instrumentation that can achieve high performance imaging not only using conventional optics, but also using additional optical excitation and collection techniques. For any new method—particularly one that champions an imaging technology that has gained traction in recent years—to make an impression in this community would require the author to clearly specify the metrics that justify the use of this technique against conventional methods. As it stands, I believe the discussion section would be strengthened by a summative statement or table that specifies clearly where metalenses have an advantage over high performance high NA objectives. E.g. lines 246 to 248 in the manuscript provide a compelling argument in favor of using metalenses due to the detectability of nanoparticles below the diffraction limit. If this is indeed an attribute that allows the lens to improve over conventional high performance objectives, this should be made clear in the discussion even if the noise associated with the structure hinders the collection of clear data. Otherwise, the metalens appears to merely act as a substitute (a chromatically limited one too) in this experiment's image relay system.

Our response:

We thank the reviewer for an elaborate summary of our work. Indeed, the main value of our work lies in the ability to reach single molecule sensitivity using a metalens system. The intense research has been directed toward improving metalens designs to achieve performances close to objective lenses in terms of either NA or focusing efficiency or chromatic aberration correction or increased size. However, optimizing all those parameters remains a challenge for the community, especially given that there are theoretical limits hindering optimization of one parameter without sacrificing the other. So, our work shows that optimizing a metalens design with a trade-off between those parameters can enable the single molecule sensitivity by using a mere one-layer micrometer-thick dielectric nanostructure arrangement. Although the metalens functionalities substitute here those of commercial objective lenses, in our opinion, this is a significant advance in the metasurface application field. Moreover, as the reviewer already mentioned the metalens platform is compact and flat which opens new avenues for making flat on-chip single-molecule sensing devices that could not be possible before due to costly and bulky objective lenses.

Now, we would like to address reviewer's comment on the comparison of the metalens to the conventional lenses. As pointed out in the responses to comment #1 of Reviewer 2, we compared

the metalens performance to the performance of aspheric lenses of NA = 0.18 and 0.54 and achromatic objective lenses of NA between 0.15 and 1.2. We record FCS data of diffusing Alexa 647 molecules. We show that the aspheric lens of NA similar to that of our metalens and has a large volume and cost doesn't allow reaching single molecule sensitivity under same excitation/detection conditions as the metalens. Our metalens exceeds the performance of objective lenses with NA < 0.5 by all critical FCS parameters such as single molecule brightness, effective FCS volume, and diffusion time of molecules. Then, the metalens-enabled single molecule brightness and diffusion time fall short to those retrieved by NA = 0.5 objective lens. The confocal effective volume appears to be similar to the conventional objective lens of a similar NA. We made a new figure (Figure 5) to emphasize the miniaturization degree and performance comparison to the conventional optics. We included the part on the metalens performance comparison against conventional optics in the main text (Pages 15 and 16).

Figure 5. Metalens performance comparison against conventional optics. (a) Set of lenses employed for comparative study with indicated size dimensions: metalens, two single-element aspheric lenses of NA = 0.18 and NA = 0.54, and achromatic objective lenses of NA between 0.15 and 1.2. (b) Alexa 647 single molecule brightness detected by metalens and aspheric lenses. As FCS correlation is not detectable by aspheric lenses, single molecule brightness is considered to equal zero. (c) Alexa 647 single molecule brightness detected by metalens and achromatic objective lenses. (d) Alexa 647 molecule diffusion time detected by metalens and achromatic objective lenses. (e) Confocal effective volume observed by metalens and achromatic objective lenses.

“In order to provide an adequate comparison between our metalens performance and the conventional optics, we record FCS data of diffusing Alexa 647 molecules by a set of lenses of

different NA and chromatic aberration presences (Figure 5a). The fluorescence from Alexa 647 solution is excited and collected by two aspheric lenses of NA = 0.18 and 0.54 and objective lenses of various NA in the range of 0.15 and 1.2 (lens sizes shown in Table S4). The corresponding raw FCS correlation traces are shown in Figure S15. We observe that the aspheric lenses of NA close to that of the metalens or below do not enable reaching single molecule sensitivity under identical excitation/detection conditions (Figure 5b). The lack of sensitivity comes from the mismatch in overlap between the excitation and detection volume of the single-element aspheric lens due to its intrinsic chromatic distortion.^{51,52} Employing achromatic multi-element objective lenses clearly boosts the single molecule fluorescence intensity (Figure 5c), although it still remains undetectable for objective lenses with NA < 0.25. The designed metalens exceeds the performance of objective lenses with NA < 0.5 by all critical FCS parameters such as single molecule brightness, effective FCS volume, and diffusion time of molecules (Figures 5c-e). Although the metalens-enabled single molecule brightness and diffusion time fall short of those retrieved by NA = 0.5 objective lens, the effective volume comes close to the conventional objective lens trend.”

The paper already describes some of the deficiencies associated with metalenses and their realization in practice. First, replicating the designed phase functions is made difficult by the precision required during nanofabrication, leading to some of the efficiency losses mentioned in the discussion section of the work. Second, their limited chromatic bandwidth inevitably necessitates the use of different lenses when operating along the other wavelengths in a fluorescence microscope's laser line--something obviated in a traditional microscope objective. This behavior is already well known in a community accustomed to the deficiencies of single surface lenses based on highly chromatic pillar designs. What should be highlighted more in the metalens-centered figures (e.g. Fig 2 and 3) is a clear correlation of how the theoretical and fabricated lenses' focusing properties benefit this experiment. Was there a better fabricated metalens with a higher focusing efficiency that can achieve detection better? Do different lens NAs provide advantages in their PSF and the ability to detect the particles under investigation? Are there other libraries that can function better than the current amorphous silicon fins?

Our response:

We have tried various specs of metalens design, including altering the lens size and adjusting its numerical aperture (NA) to apply on FCS experiment. As shown in figure below, we present results for lenses with a size of 1.1 μm , specifically at two different focal lengths: 150 (sample 1) and 330 (sample 2).

Rvs Figure 3. The result of other fabricated samples. (a) OM images of surface of sample metalens 1. (b) Brief focusing test of sample metalens 1. (c) OM images of surface of sample metalens 2. (d) Brief focusing test of sample metalens 2.

As evident from the results, the fabrication quality of the lenses was suboptimal, resulting in less favorable focusing shapes that exhibited a rectangular pattern. This led to a lower quality PSF, rendering them unsuitable for FCS experiments. The metalens used in manuscript represented the highest quality PSF among several lenses we designed and fabricated. And with this lens, nanoparticle detection experiment was done successfully.

In Prof. Junsuk Rho's laboratory at POSTECH, there is a library that allows for the design of optimal materials depending on the operating wavelength. For example, materials such as TiO_2 (Yoon et al., *Nature Communications*, 11.1 (2020): 2268) were employed to operate in the entire visible light spectrum, and ZrO_2 (Kim et al., *Light: Science & Applications*, 12.1 (2023): 68) was utilized for operation in the UV region. These papers have been recently published in scientific journals.

In the context of this investigation, amorphous silicon was selected due to its notable transmittance characteristics within the red spectral region. It is important to note that SiN or crystalline Si could potentially serve as viable alternatives to amorphous silicon. The current fabrication conditions are specifically tailored to amorphous silicon. Therefore, should there be an intention to utilize alternative materials, a diligent effort would be required to identify and establish the optimal fabrication conditions tailored to these alternatives.

I do appreciate the work that went into capture the actual lens' PSF--the FWHM demonstrates the feasibility of applying such a structures towards detecting some of the smaller structures investigated in this study (like the Alexa dye). However, I wish that Figures 2 and 3 were better substantiated with information like this that really rationalizes the power of using these high NA flat optics.

Our response:

We thank the reviewer for the suggestion. We have modified Figure 2 (Page 7 in the main text) to highlight the importance of employing a metalens featuring achromaticity and high focusing efficiency and NA for single molecule sensing. We removed the figure panels showing the meta-atom geometry arrangements (phase maps) and included a schematic of the overlapping excitation and detection volumes of the metalens and the molecule detection efficiency (MDE). As pointed out by refs.51 and 52, the mismatch in overlap of the excitation and detection volumes leads to the elongation of the FCS effective volume, and at extreme cases to the loss of single molecule brightness even for high NA lenses. The schematic shows the match of the volume positions for excitation and emission maximum to minimize this problem. On the other hand, the molecule detection efficiency is a product of the laser power density, *i.e.* the rate of pumping the molecules to their excited states, and the collection efficiency of the system (CEF). The laser power density is proportional to FE_{exc}/ω_0^2 with FE_{exc} being the metalens focusing efficiency (transmittance in the case of conventional objective lens) at the laser wavelength. Hence, under the diffraction limit approximation the power density of excitation light becomes proportional to $FE_{exc}\cdot NA^2$. On the other hand, CEF is proportional to $FE_{em}(1 - \sqrt{1 - (NA/n)^2})$, where FE_{em} is the metalens focusing efficiency (transmittance in the case of conventional objective lens) at the emission wavelength, n is the refractive index of the aqueous solution which amounts to 1.33. We show the MDE plot as a function of NA at four exemplary focusing efficiencies of metalens assuming them equal for both wavelengths. Clearly, both NA and focusing efficiency drastically affect MDE, therefore have to be obtained optimized together. We also add a discussion on the importance of those parameters in the main text (Page 9).

On the other hand, we kindly would like to express our confidence that Figure 3 carries important experimental verification that the metalens has a chance to excite and properly collect single molecule fluorescence, and that the fabrication are implemented with high quality. Although, as we indicated in the previous comments, we added new Figure 5 to provide a full comparison of the flat metalens performance and size with conventional optics to highlight the appropriateness and power of using such flat optics.

Figure 2. Metalens design. (a) Bright-field optical microscope image of fabricated metalens. (b) and (c) Scanning electron microscope images of metalens in center and away from center, respectively. (d) Schematic of dual-wavelength metalens functionality. Overlap between excitation and detection volume minimizes confocal effective volume and provides efficient light collection. (e) Calculated molecule detection efficiency as function of lens NA and focusing efficiency (FE). Molecule detection efficiency is represented in arbitrary units, as its absolute values strictly depend on total transmission of optical microscope setup. FEs at excitation and collection wavelengths are considered identical for this schematic.

“A representative optical microscopy image of the metalens is shown in Figure 2a. According to the scanning electron microscope images, the fabricated metalens accommodates well-defined rectangular meta-atoms with sharp edges (Figures 2b and 2c). The rationale behind the optimization of focusing efficiency, NA, and chromatic aberration correction takes into consideration two main aspects: overlap of excitation and detection volumes (Figure 2d) and molecule detection efficiency (MDE). The overlap mismatch of the excitation and detection volumes leads to the elongation of the FCS effective volume and limited single molecule brightness even for high NA lenses.⁵¹ MDE represents a metric of single molecule fluorescence intensity the system can detect and amounts to a product of the laser power density, *i.e.* the rate of pumping the molecules to their singlet excited state, and the collection efficiency of the system (CEF).⁵² Under the diffraction limit conditions, the laser power density is proportional to $FE_{exc} \cdot NA^2$ with FE_{exc} being the metalens focusing efficiency (transmittance in case of conventional objective lens) at the laser wavelength. On the other hand, for a correctly set pinhole size, CEF is proportional to $FE_{em} (1 - \sqrt{1 - (NA/n)^2})$,⁵² where FE_{em} is the metalens focusing efficiency (transmittance in case of conventional objective lens) at the emission wavelength, n is the refractive index of the aqueous solution. Figure 2e displays the MDE plot as a function of NA at

four exemplary metalens focusing efficiencies. Both NA and focusing efficiency drastically affect MDE, therefore, our rationale for a fabricable metalens design accommodates their trade-off together with dual-wavelength operation.”

I also appreciate the volume of data collection that went into completing Figures 4 and 5. While it suggests flaws in the nature of the hardware when compared against standard equipment (especially due to the flaws of the fabricated nanostructures), it validates the use case depicted by this paper and shows the feasibility of this application should the fundamental issues of metalens fabrication and performance be resolved in future designs.

Our response:

As the reviewer commented, advancements in metasurface design and fabrication standards have the potential to elevate the single molecule brightness enabled by the metalens platform. As illustrated in the figure below, we are currently facing challenges related to slight variations in the nanostructure shapes based on fabrication conditions. Furthermore, there are instances where the structural shape in the vertical direction undergoes changes depending on the position of nanostructure during the etching process.

Rvs Figure 4. Top view analysis: the fabricated sample in (a) condition 1 and in (b) condition 2. (c) Oblique view analysis.

Consequently, we are dedicated to designing metalenses that can function adequately even with minor alterations in structure. To address this, we are particularly focused on predicting structural changes that occur during the fabrication process. We believe that by addressing these challenges effectively, we can mitigate potential performance degradation in meta-lenses.

I'd like to request the author to include a stronger summative statement with a table that enumerates the strengths of the metasurface based imaging system described in this paper against the conventional approach. It is inevitably going to suffer in performance when compared against traditional fluorescence microscopy, but these metrics need to be clearly laid out so that the researchers who follow up on this line of work can clearly reference the benchmark demonstrated here.

Our response:

We thank the reviewer for this comment. We kindly refer the reviewer to our response to Comment 1 and to Reviewer #2 Comment 1 regarding the metalens characteristics for single molecule sensing and comparisons with performance of conventional optics. We provided an elaborate comparison of metalens performance against aspherical single-element lenses and achromatic objective lenses. New Figure 5 shows all the main metrics derived from the FCS approach. Single molecule brightness is a metric of molecule detection efficiency. Molecule diffusion time is a metric of focusing sharpness. Lastly, effective volume is a metric of ensemble of chromatic distortion, focusing sharpness and depth of field. Thus, we highlight that metalens can perform better than single-element aspheric lenses or low NA objective lenses. Additionally, this figure emphasizes the miniaturization of the metalens platform compared to conventional optics. Also, we also included a table with exact sizes of the metalens and the conventional optics objects as below (Supplementary Information section S15, Table S4).

Table S4. Metalens size comparison against conventional optics.

Lens type	Numerical aperture	Height (mm)	Width (mm)
Metalens	0.6	0.5·10 ⁻³ (structure) 0.5 (substrate)	0.5
Aspheric lenses	0.18	3	7.5
	0.54	9	25
Achromatic objective lenses	0.15	30	25
	0.25	40	25
	0.3	40	25
	0.5	50	25
	0.8	50	25
	0.95	65	33
	1.2	65	34.5

A softer request is to substantiate Figures 2 and 3 with metalens performance metrics that more directly correlate to the function of the experiment at hand. Phase functions are phase functions-how well you meet them should be reflected by the efficiency and ought to be more of a supporting information kind of inclusion. Is there data that you already have in the SI that can better reflect your unique metalens' designs benefits for FCS and single molecule detection?

Our response:

We agree with the reviewer that the maps of the meta-atom widths and lengths appear more appropriate to the Supplementary Information. Therefore, as we mentioned in our response to comment #3, we have modified Figure 2 by removing the panels related to meta-atom and meta-atom geometry maps, and included more information regarding the design consideration. We pointed out now the consideration of two-wavelength focusing to maintain match in overlap of the excitation and detection volumes and the molecule detection efficiency as discussed in previous responses. The latter parameter is clearly dependent on both NA and focusing efficiency. Modified Figure 2 looks as below.

Figure 2. Metalens design. (a) Bright-field optical microscope image of fabricated metalens. (b) and (c) Scanning electron microscope images of metalens in center and away from center, respectively. (d) Schematic of dual-wavelength metalens functionality. Overlap between excitation and detection volume minimizes confocal effective volume and provides efficient light collection. (e) Calculated molecule detection efficiency as function of lens NA and focusing efficiency (FE). Molecule detection efficiency is represented in arbitrary units, as its absolute values strictly depend on total transmission of optical microscope setup. FEs at excitation and collection wavelengths are considered identical for this schematic.

To reach those requirements as well as technical points associated with confocal microscopy, the metalens design parameters had to satisfy the following parameters:

1. Working distance (focal length) > 200 μm (comfortable distance to focus through confocal glass slide)
2. High NA ≥ 0.6 which resulted in diameter ~500 μm
3. Working wavelength can cover the excitation and collection wavelengths of fluorescent beads. (635 nm & 670 nm)
4. Average transmittance > 85 %

5. Polarization-insensitivity: metalens can working in all polarization condition

We have selected the most effective method from the available options (using co-polarization vs cross-polarization). The figure presented below provides a visual representation of the outcomes when alternative phase modulation methods are employed. The conversion efficiencies by using co-polarization term are larger than the efficiencies by using cross-polarization term.

Figure S7. Comparing conversion efficiencies between co-pol and cross-pol. (a) Conversion efficiency at 635-nm incidence by using co-polarization term. (b) Conversion efficiency at 670-nm incidence by using co-polarization term. (c) Conversion efficiency at 635-nm incidence by using cross-polarization term. (d) Conversion efficiency at 670-nm incidence by using cross-polarization term.

Through the utilization of this dual-focusing metalens, we have effectively attained precise focusing and established its practical utility in FCS experiments. We have included these results in the Supplementary Information and provided corresponding explanations in the manuscript (Pages 8 and 9) as below.

“This result represents the conversion efficiency obtained when using the co-polarization term, which is more than 5% higher than the conversion efficiency observed when using cross-polarization (Figure S7).”

These small changes aside, I believe this paper has value to add to the literature for the novel photonics and molecule sensing/device space.

Our response:

We deeply appreciate the reviewer’s recognition of the value of our work.

REVIEWERS' COMMENTS:

Reviewer #1 (Remarks to the Author):

I believe that the authors have satisfactorily addressed the reviewers' comments, and I recommend acceptance.

Reviewer #2 (Remarks to the Author):

The authors revised their manuscript and took some of my comments on-board. Most of my comments agreed with Reviewer's 3, where I wanted the novelty of the work to be better demonstrated and discussed.

The authors have improved on this point a bit. Ideally, I would have liked to see more, and if the authors get the chance I would advise them to do so.

For example more clear, impactful statements like: "Compared to a conventional microscope with similar properties, we managed to miniaturize the set-up by ..%. Of course one can work and improve the fabrication and performance of the metalens to achieve current state-of-the-art measurements, and even surpass them, with such a miniaturized"

I believe the work presented will be better received and attract more attention (and citations) if it is more clear on how it compares with current techniques.

Nevertheless, if the rest of the reviewers are happy to accept this manuscript for publication, I am happy to also accept it for publication in its current form.

Reviewer #3 (Remarks to the Author):

Changes to the text enabled by the exchange with the peer reviewers has significantly improved the figures and the discussion in both the manuscript's main body and the supporting information. I believe that the improved figures and text justify the publication of this paper in Nature Communications. Not only is the metalens design and nano fabrication aspect addressed more thoroughly in this revision, but the new and revised

figures improve the readers ability to understand this work's technological impact in the bio imaging space.

I recommend this work's publication in Nature Communications.

Reviewer #1 (Remarks to the Author):

I believe that the authors have satisfactorily addressed the reviewers' comments, and I recommend acceptance.

Our response:

We are grateful to the reviewer for his/her time in handling our manuscript and thank the reviewer for his/her favorable decision.

Reviewer #2 (Remarks to the Author):

The authors revised their manuscript and took some of my comments on-board. Most of my comments agreed with Reviewer's 3, where I wanted the novelty of the work to be better demonstrated and discussed.

The authors have improved on this point a bit. Ideally, I would have liked to see more, and if the authors get the chance I would advice them to do so.

For example more clear, impactful statements like: "Compared to a conventional microscope with similar properties, we managed to minuatirize the set-up by ..%. Of course one can work and improve the fabrication and performace of the metalens to achieve current state-of-the-art measurements, and even surpass them, with such a miniaturized"

I believe the work presented will be better received and attract more attention (and citations) if it is more clear on how it compares with current techniques.

Our response:

We agree with the important remark from the reviewer. We added several comments in the main text of the manuscript highlighting the novelty of our work and the advantages of the proposed methodology as compared to conventional microscope systems. We currently state that the metalens miniaturizes the objective lens, a key optical element for single molecule sensing, by around 2 orders of magnitude in all dimensions, whereas it maintains the single molecule sensitivity. We also highlight that the single-layer metalens achieves single molecule sensitivity, while the performance of refractive single-element refractive lenses is typically flawed even at high NA. The known abilities of metalenses to have tunability of electromagnetic field manipulation as a function of, e.g., polarization, wavelength, orbital angular momentum, could provide compact platforms for light manipulation and sorting, which could be beneficial to remove other conventional optical elements from experimental detection systems, such as beam splitters and spectral filters. Lastly, we highlight the fact that the silicon metalens is compatible with photonic integrated circuits which could potentially open ways to create single-molecule systems-on-chip of ultimate portability. The comments are added to the manuscript on Pages 5, 17, 18, 20, 21 (Track-changes version) as below:

“Compared to conventional objective lenses with similar properties, the designed metalens yields comparable single molecule sensitivity with a single layer of dielectric nanofins, while the lens size is miniaturized by around 2 orders of magnitude in all dimensions.”

“Altogether, the proposed metalens device miniaturizes conventional objective lens size by 2 orders of magnitude in all dimensions and maintains similar single molecule sensitivity. Moreover, the demonstrated dual-wavelength operation of the metalens encoded within a single nanostructure layer plays a pivotal role in providing sufficient collection efficiency from diffusing molecules, whereas conventional single-element refractive lenses typically strive to collect sufficient single molecule fluorescence signal even at high NA⁵².”

“The demonstrated performance already allows considerable miniaturization and substitution of costly objective lenses of modest NA. Considering that future improvements in metalens fabrication and performance could achieve the single molecule sensitivity of current conventional state-of-the-art measurements and enable additional functionalities with unprecedented control of light properties. For instance, the unique tunability of electromagnetic field manipulation by metasurfaces^{64–66} opens horizons for future developments in single molecule spectroscopy, such as directionality manipulation and sorting of emission or additional system miniaturization of multicolor detection systems by substituting functionalities of beam splitters and spectral filters. Another advantage of the proposed methodology includes its potential integration with miniaturized microscopes or photonic integrated circuits for handheld single molecule sensors with ultimate miniaturization and portability.”

Nevertheless, if the rest of the reviewers are happy to accept this manuscript for publication, I am happy to also accept it for publication in its current form.

Our response:

We appreciate the reviewer’s approval to accept our work for publication.

Reviewer #3 (Remarks to the Author):

Changes to the text enabled by the exchange with the peer reviewers has significantly improved the figures and the discussion in both the manuscript's main body and the supporting information. I believe that the improved figures and text justify the publication of this paper in Nature Communications. Not only is the metalens design and nano fabrication aspect addressed more thoroughly in this revision, but the new and revised figures improve the readers ability to understand this work's technological impact in the bio imaging space.

I recommend this work's publication in Nature Communications.

Our response:

We deeply appreciate the reviewer’s recognition of our work’s impact and the recommendation of the manuscript publication.